# Intermittent PI3Kδ inhibition sustains anti-tumour immunity and curbs irAEs

Simon Eschweiler[1], Ciro Ramírez-Suástegui[1], Yingcong Li[1,2], Emma King[3,4], Lindsey Chudley[5], Jaya Thomas[3], Oliver Wood[3], Adrian von Witzleben[3,6], Danielle Jeffrey[3], Katy McCann[3], Hayley Simon[1], Monalisa Mondal[1], Alice Wang[1], Martina Dicker[1], Elena Lopez-Guadamillas[7], Ting-Fang Chou[1], Nicola A. Dobbs[8], Louisa Essame[8], Gary Acton[8], Fiona Kelly[8], Gavin Halbert[9], Joseph J. Sacco[5,10], Andrew Graeme Schache[5,11], Richard Shaw[5,11], James Anthony McCaul[12], Claire Paterson[13], Joseph H. Davies[4], Peter A. Brennan[14], Rabindra P. Singh[15], Paul M. Loadman[16], William Wilson[17], Allan Hackshaw[17], Gregory Seumois[1], Klaus Okkenhaug[18], Gareth J. Thomas[3], Terry M. Jones[5,11], Ferhat Ay[1], Greg Friberg[19], Mitchell Kronenberg[1,2], Bart Vanhaesebroeck[7], Pandurangan Vijayanand[1,5,20,21✉] & Christian H. Ottensmeier[1,3,5,10,21✉]

Phosphoinositide 3-kinase δ (PI3Kδ) has a key role in lymphocytes, and inhibitors that target this PI3K have been approved for treatment of B cell malignancies[1–3]. Although studies in mouse models of solid tumours have demonstrated that PI3Kδ inhibitors (PI3Kδi) can induce anti-tumour immunity[4,5], its effect on solid tumours in humans remains unclear. Here we assessed the effects of the PI3Kδi AMG319 in human patients with head and neck cancer in a neoadjuvant, double-blind, placebo-controlled randomized phase II trial (EudraCT no. 2014-004388-20). PI3Kδ inhibition decreased the number of tumour-infiltrating regulatory T ($T_{reg}$) cells and enhanced the cytotoxic potential of tumour-infiltrating T cells. At the tested doses of AMG319, immune-related adverse events (irAEs) required treatment to be discontinued in 12 out of 21 of patients treated with AMG319, suggestive of systemic effects on $T_{reg}$ cells. Accordingly, in mouse models, PI3Kδi decreased the number of $T_{reg}$ cells systemically and caused colitis. Single-cell RNA-sequencing analysis revealed a PI3Kδi-driven loss of tissue-resident colonic ST2 $T_{reg}$ cells, accompanied by expansion of pathogenic T helper 17 ($T_H17$) and type 17 $CD8^+$ T ($T_C17$) cells, which probably contributed to toxicity; this points towards a specific mode of action for the emergence of irAEs. A modified treatment regimen with intermittent dosing of PI3Kδi in mouse models led to a significant decrease in tumour growth without inducing pathogenic T cells in colonic tissue, indicating that alternative dosing regimens might limit toxicity.

PI3K inhibitors were initially considered to target mainly PI3K activity intrinsic to cancer cells, which was the underlying rationale for testing inhibitors against the leukocyte-enriched PI3Kδ in B cell malignancies. However, subsequent studies have shown that PI3Kδ inhibition also has clear immunomodulatory activities, largely T cell-mediated, that were under-appreciated at the time of the early trials in B cell malignancies, causing irAEs that have hampered clinical progress and utility. Several lines of evidence suggest that PI3Kδi preferentially inhibit $T_{reg}$ cells over other T cell subsets[4–8] but so far, no trials have been performed to explicitly explore this concept in humans. Here we provide an in-depth investigation of the effect of PI3Kδ inhibition on immune cells in patients with solid tumours and also explore the mechanism that leads to irAEs.

## PI3Kδ inhibition causes irAEs

To evaluate the potential of PI3Kδi as immunotherapeutic agents in human solid cancers, we administered the PI3Kδi AMG319 to

[1]La Jolla Institute for Immunology, La Jolla, CA, USA. [2]Division of Biological Sciences, University of California San Diego, La Jolla, CA, USA. [3]CRUK and NIHR Experimental Cancer Medicine Center, University of Southampton, Southampton, UK. [4]Dorset Cancer Centre, Poole Hospital NHS Foundation Trust, Poole, UK. [5]Liverpool Head and Neck Center and Institute of Systems, Molecular and Integrative Biology, University of Liverpool, Liverpool, UK. [6]Department of Otorhinolaryngology, Head and Neck Surgery, Ulm University Medical Center, Ulm, Germany. [7]UCL Cancer Institute, University College London, London, UK. [8]Centre for Drug Development, Cancer Research UK, London, UK. [9]Cancer Research UK Formulation Unit, University of Strathclyde, Glasgow, UK. [10]Clatterbridge Cancer Centre NHS Foundation Trust and Liverpool Cancer Research UK Experimental Cancer Medicine Center Liverpool, Liverpool, UK. [11]Liverpool University Hospitals NHS Foundation Trust, Liverpool, UK. [12]Queen Elizabeth University Hospital, Glasgow, UK. [13]Beatson West of Scotland Cancer Centre, Glasgow, UK. [14]Queen Alexandra Hospital, Portsmouth, UK. [15]Southampton University Hospitals NHS Foundation Trust, Southampton, UK. [16]University of Bradford, Institute of Cancer Therapeutics, Bradford, UK. [17]Cancer Research UK and UCL Cancer Trials Centre, London, UK. [18]Department of Pathology, University of Cambridge, Cambridge, UK. [19]Amgen, Thousand Oaks, CA, USA. [20]Department of Medicine, University of California San Diego, La Jolla, CA, USA. [21]These authors jointly supervised this work: Pandurangan Vijayanand, Christian H. Ottensmeier. ✉e-mail: vijay@lji.org; c.ottensmeier@liverpool.ac.uk

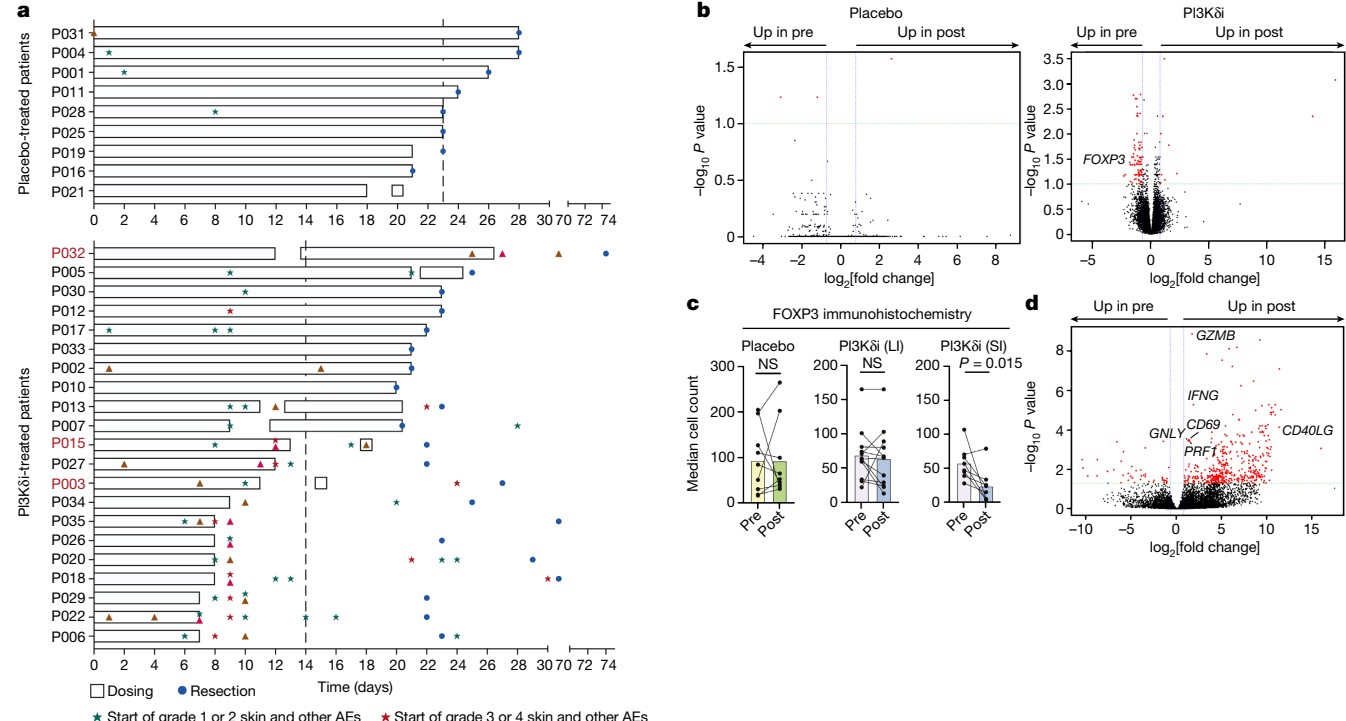

**Fig. 1 | PI3Kδi drives anti-tumour immunity but causes significant irAEs.**
**a**, Swimlane plot depicting treatment regimen, intervals and occurrence and
grade of irAEs in PI3Kδi-treated (top) and placebo-treated (bottom) patients;
patients with partial response or complete pathological response are
highlighted in magenta. Vertical dashed lines show average duration of
treatment. **b**, **d**, Volcano plots of whole-tumour RNA-seq analysis (**b**) or bulk
RNA-seq analysis of purified tumour-infiltrating CD8⁺ T cells (**d**) comparing
patients treated with AMG-319 to those treated with placebo. DEGs between
pre- and post-treatment samples are highlighted in red and were called by
DEseq2; adjusted *P* values were calculated with the Benjamini–Hochberg

method. Depicted are transcripts that changed in expression by more than
0.75-fold and had an adjusted *P* value of ≤0.1 (**b**) or <0.05 (**d**). **c**, Median cell
count of FOXP3⁺ cells in pre- and post-treatment samples of placebo- or
AMG319-treated patients. AMG-319-treated patients have been further
stratified into patients for whom the interval between stopping of treatment
and immunohistochemistry assessment was more than four days (long interval
(LI)) or less than one day (short interval (SI)). *P* = 0.015 for SI. Data are
mean ± s.e.m.; two-tailed Wilcoxon matched-pairs signed rank test
(**c**). Differential expression analysis (**b**, **d**) was performed using DESeq2
(v1.24.0). AE, adverse event; NS, not significant.

treatment-naive patients with resectable head and neck squamous cell
carcinoma (HNSCC) in a neoadjuvant, double-blind, placebo-controlled
randomized phase II trial (Extended Data Fig. 1a, b). We measured tar-
get inhibition (using phosphorylated AKT (pAKT) levels in B cells)
(Extended Data Fig. 1c) and drug levels (Extended Data Fig. 1d) to verify
drug administration. Thirty-three patients were randomized in a 2:1
ratio (AMG319:placebo) for the trial and 30 patients received at least
one dose of AMG319 or placebo. Fifteen patients received 400 mg
daily of AMG319 (range of 7–24 days per patient). Unexpectedly, at the
400 mg dose, 9 out of 15 patients experienced irAEs that led to with-
drawal of treatment. After a formal safety review, 6 additional patients
were recruited and treated at a reduced dose of 300 mg per day. Again,
three out of six patients had irAEs that led to discontinuation of treat-
ment. One patient experienced grade 4 colitis after completion of 24
daily doses of AMG319 and eventually required colectomy (Fig. 1a).
The most prevalent irAEs were skin rashes (29%; 25% in the treatment
group and 4% in placebo group), diarrhea (29%; 28% in the treatment
group and 1% in placebo group) and transaminitis (14% all in the treat-
ment group), consistent with a treatment-mediated loss of T_reg cells or
T_reg cell functionality in multiple tissues causing immunopathology.
The onset of irAEs was surprisingly rapid (median time to onset of 9 days)
and led to treatment discontinuation in 12 out of 21 AMG319-treated
patients. Clinically, and most probably reflecting the brief treatment
period, we did not observe any significant differences in the measured
tumour volumes between the study arms in the 23 patients in whom
this was evaluable. Two patients with partial responses and one with

complete pathological response were among the AMG319-treated
patients (Extended Data Fig. 1c), all of whom also exhibited grade 3/4
irAEs.

## PI3Kδi alter the tumour microenvironment

Whole-tumour RNA-sequencing (RNA-seq) analysis of pre- and
post-treatment tumour samples revealed substantial differences
in the AMG319 treatment group (93 differentially expressed genes
(DEGs)), but not for the placebo group (3 DEGs) (Fig. 1b). As PI3Kδ inhi-
bition led to a significant reduction in *FOXP3* transcript levels in the
tumour samples (Fig. 1b), we assessed T_reg cell levels in tumour tissue
by immunohistochemistry, hypothesizing that the duration between
cessation of treatment and tumour resection might be a critical factor
influencing T_reg cell abundance, owing to the relatively short half-life of
the compound. Indeed, we found significantly reduced intratumoural
T_reg cells only in patients in which their abundance could be assessed
directly after treatment (PI3Kδi short interval) (Fig. 1c), implying that
T_reg cell levels normalize quickly once treatment has been stopped.

Bulk RNA-seq analysis of sorted tumour-infiltrating CD8⁺ T cells
revealed higher expression of *IFNG*, *GZMB* and *PRF1* in post-treatment
samples, indicating enhanced cytotoxic potential of tumour-infiltrating
CD8⁺ T cells following PI3Kδi treatment (Fig. 1d). We corroborated these
results by single-cell RNA-seq (scRNA-seq) analysis, which demon-
strated that CD4⁺ and CD8⁺ T cell clusters showed a treatment-associated
increase in expression of cytotoxicity genes (for example, *GZMB* and

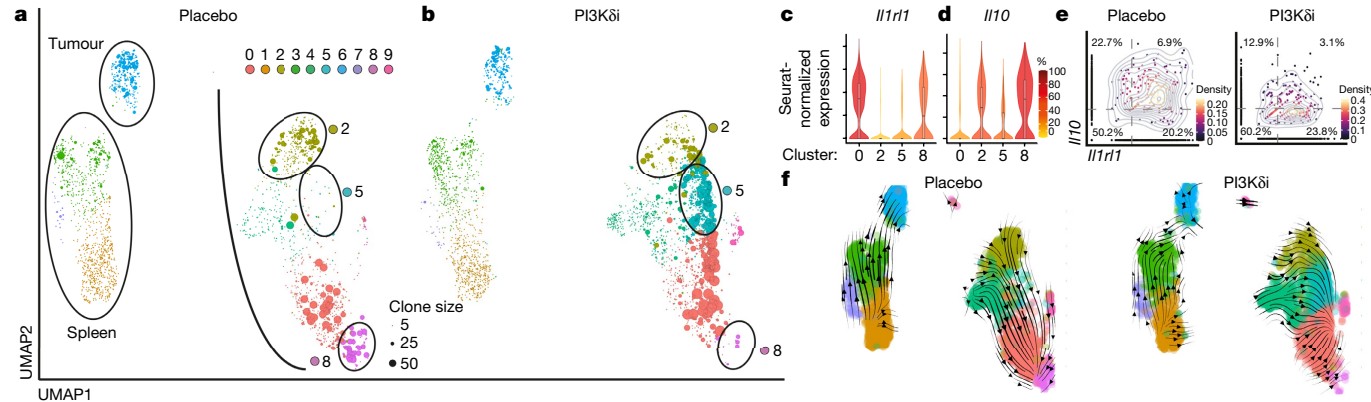

**Fig. 2 | PI3Kδi affects distinct T_reg cell subtypes. a, b,** Uniform manifold approximation and projection (UMAP) plots single-cell transcriptomes and T cell receptor (TCR) sequence data of FOXP3⁺CD4⁺ T cells in placebo-treated control mice (**a**; *n* = 3 mice) and PI-3065-treated mice (**b**; *n* = 3 mice). Circle size indicates degree of clonal expansion. **c, d,** Violin plots showing Seurat-normalized expression levels (colour scale depicts percentage of expressing cells) of highlighted genes in the indicated clusters from **a, b**. The centre line depicts the median, edges delineate the 25th and 75th percentiles and whiskers extend to minimum and maximum values. **e,** Scatter plots showing Seurat-normalized expression levels of highlighted genes in colonic T_reg cells in placebo-treated and PI-3065-treated mice. The dashed line indicates the expression cut-off; numbers indicate the frequency in each quadrant. **f,** RNA velocity analysis visualized by UMAP, depicting likely developmental trajectories of T_reg cells from **a, b**. Arrows indicate velocity streamlines.

*PRF1*) (Extended Data Fig. 2a, b). We also found a modest clonal expansion of CD4⁺ and CD8⁺ T cells after treatment (Extended Data Fig. 2c). As low cell numbers of CD4⁺FOXP3⁺ T cells (0–27 cells per patient) precluded a more detailed analysis in our cohort, we next assessed circulating T_reg cells. PI3Kδ inhibition led to a significant increase in activated circulating T_reg cells, while the proportion and activation status of T_reg cells in the placebo group remained stable (Extended Data Fig. 2d, e). This implies that PI3Kδ inhibition either influences proliferation or displaces activated T_reg cells from tissues, presumably by altering the expression of tissue homing factors such as KLF2 and S1PR1 (direct targets of FOXO1 in line with previous studies[5–7]), probably contributing to toxicity. Together, these data indicate that PI3Kδ inhibition causes profound changes in the tumour microenvironment (TME), characterized by enhanced CD4⁺ and CD8⁺ T cell activation, oligoclonal T cell expansion and increased cytolytic activity, consistent with a decrease in intratumoural T_reg cells, enabling T cell activation and leading to a rapid onset of dose-limiting toxicity.

## Systemic effects of PI3Kδi on T_reg cells

Next, to understand the mechanistic basis of PI3Kδi-induced toxicity and anti-tumour immune responses, we tested the effect of a PI3Kδ inhibitor in a mouse solid tumour model. We inoculated wild-type C57BL/6 mice with B16F10-OVA melanoma cells and treated them with the PI3Kδi PI-3065[7]. Consistent with previous studies[4,5], we found a significant decrease in tumour volume (Extended Data Fig. 3a) and a significant increase in the number of intratumoural CD8⁺ T cells that expressed high levels of PD-1 and exhibited increased proliferative and cytotoxic capacity (Extended Data Fig. 3b–e). TOX, a transcription factor recently identified as critical for the adaptation and survival of CD8⁺ T cells in the TME[9], was also increased after PI3Kδi treatment (Extended Data Fig. 3f). Notably, and contrary to previous reports[10,11], we found that the expression of both granzyme B and Ki67 was almost exclusively limited to TOX⁺CD8⁺ T cells (Extended Data Fig. 3g), demonstrating that these cells, despite showing high expression of PD-1 and TOX, are not functionally exhausted in this tumour model.

Given that PI3K inhibitors were initially considered to target mainly cancer cell-intrinsic PI3K activity, we used *Rag1*⁻/⁻ and *Cd8*⁻/⁻ mice to verify that the observed anti-tumour effects were dependent on immune cells and, more specifically, on CD8⁺ T cells (Extended Data Fig. 4h). As PI3Kδ inhibition caused substantial irAEs in non-malignant organs (Fig. 1b), and given that T_reg cells have been shown to be susceptible to this form of treatment, we next assessed whether PI3Kδi act locally within the tumour tissue or systemically. Of note, in PI3Kδi-treated mice, but not placebo-treated control mice, we found a significant decrease in T_reg cells in tumour, spleen and colon, indicative of systemic effects of PI3Kδi on T_reg maintenance or survival (Extended Data Fig. 3i–k).

## PI3Kδi affect specific T_reg cell subsets

As gastrointestinal toxicity is one of the major irAEs in patients receiving PI3Kδi[4,6,12] (Fig. 1b), we hypothesized that T_reg cells present in colonic tissue may be especially sensitive to PI3Kδi. To test this hypothesis in an unbiased manner, we performed scRNA-seq of T_reg cells isolated from tumour, spleen (lymphoid organ) and colonic tissue of PI3Kδi- and placebo-treated B16F10-OVA tumour-bearing mice. UMAP analysis identified 10 T_reg cell clusters, implying substantial T_reg cell heterogeneity and tissue-dependent adaptations (Fig. 2a, b); this supports the notion that several distinct T_reg subtypes exist in different locations (Extended Data Fig. 4a, b), in agreement with previous studies[13,14]. Colonic T_reg cells exhibited the most pronounced differences between PI3Kδi and placebo treatment, with 869 DEGs, whereas splenic and tumour T_reg cells exhibited fewer differences (Extended Data Fig. 4c–e). Two of the colonic T_reg subsets (clusters 2 and 8) were depleted in PI3Kδi-treated mice (Fig. 2a, b, Extended Data Fig. 4f). Cluster 2 colonic T_reg cells were enriched for the expression of *Ctla4* and genes encoding chemokine receptors (*Ccr1*, *Ccr2* and *Ccr4*), which are critical for their suppressive[15,16] and migratory[17] capacities, respectively (Extended Data Fig. 4g). Cluster 8 colonic T_reg cells—which showed substantial clonal expansion in control-treated mice but were depleted in PI3Kδi-treated mice—resembled the recently described tissue-resident ST2 T_reg cells[18–20], which are critical for protection against chronic inflammation and facilitation of tissue repair (Fig. 2a, b). Accordingly, we found enrichment in the expression of the ST2 T_reg signature genes *Il1rl1* (which encodes IL1RL1 (also known as IL-33R or ST2)), *Gata3* and *Id2*, as well as several genes associated with highly suppressive effector T_reg cells (*Klrg1*, *Cd44*, *Cd69*, *Pdcd1*, *Areg*, *Nr4a1*, *Il10* and *Tgfb1*) (Extended Data

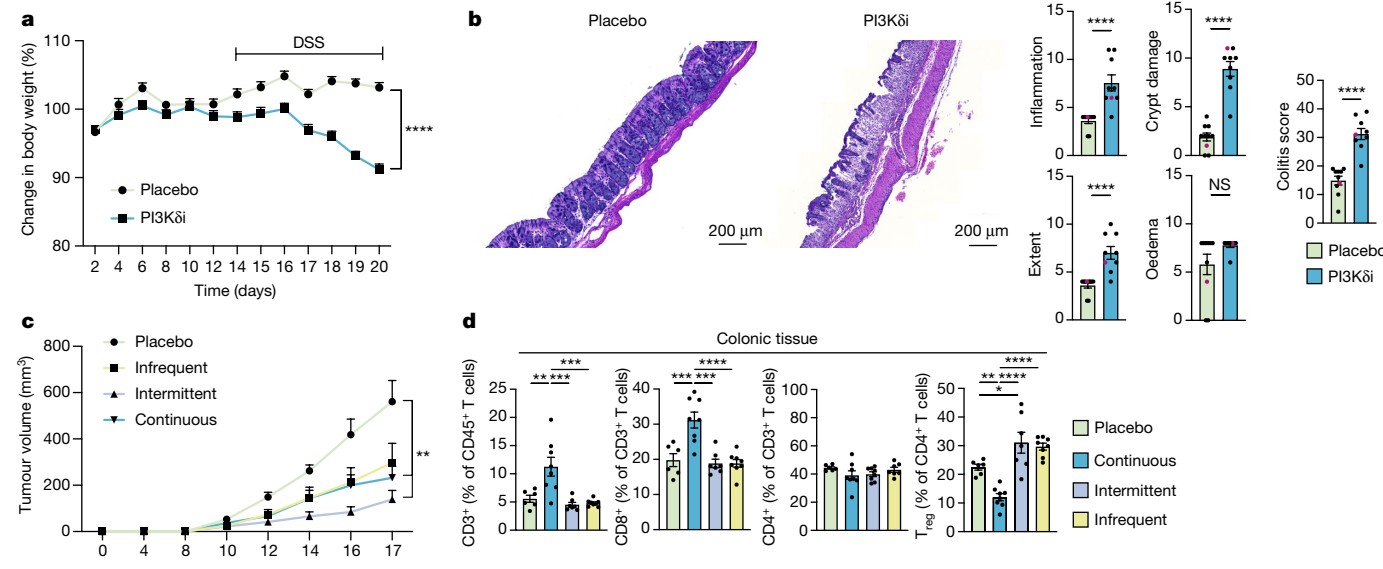

**Fig. 3 | PI3Kδi exacerbates colitis. a**, Mice were fed either a control diet or a diet containing PI-3065 for the duration of the experiment and were additionally treated with 2.5% dextran sulfate sodium (DSS) from day 14 to day 20. Change in body weight is shown relative to body weight before treatment on day 0. *n* = 10 mice per group, *P* < 0.0001. **b**, Representative sections from haematoxylin and eosin (H&E) histology scans and colitis scoring from zinc-formalin-fixed colonic tissue sections from mice treated with placebo or PI3Kδi in **a**. *n* = 10 mice (placebo) and *n* = 9 (PI3Kδi) (one mouse died before the experimental endpoint); *P* < 0.0001 for inflammation, extent, crypt damage and overall colitis scoring; representative samples from the H&E staining are highlighted in magenta. **c**, **d**, Mice were inoculated subcutaneously with B16F10-OVA cells and fed either a control diet or a diet containing PI-3065. Infrequent dosing, PI3Kδi for 2 days followed by 5 days off drug; intermittent dosing, PI3Kδi for 4 days followed by 3 days off drug; continuous dosing, PI3Kδi for the duration of the experiment. Tumour volume (**c**) and flow-cytometric

analyses of cell frequencies (**d**) of mice treated as indicated. *n* = 6 mice (placebo), *n* = 7 mice (intermittent dosing), *n* = 8 mice (continuous dosing) and *n* = 8 mice (infrequent dosing). Placebo versus intermittent dosing (**c**), *P* = 0.0023; placebo versus continuous dosing (**c**), *P* = 0.0059; placebo versus continuous dosing, *P* = 0.003; continuous dosing versus intermittent dosing and infrequent dosing (left), *P* = 0.0003; placebo versus continuous dosing, *P* = 0.0005; for continuous dosing versus intermittent dosing, *P* = 0.0001; placebo versus infrequent dosing (**d**; third from left), *P* < 0.0001; placebo versus continuous dosing, *P* = 0.0086; placebo versus intermittent dosing, *P* = 0.045; continuous dosing versus intermittent dosing and infrequent dosing, *P* < 0.0001. Data are mean ± s.e.m.; two-tailed Mann–Whitney test (**a**–**c**) or one-way ANOVA comparing the mean of each group with the mean of each other group followed by Dunnett's test (**d**). Data are representative of at least two independent experiments.

Fig. 5a). We verified ST2 expression on $T_{reg}$ cells at the protein level and found that PI3Kδ inhibition led to a substantially increased ratio of CD8[+] T cells to ST2 $T_{reg}$ cells (Extended Data Fig. 5b). Whereas colonic $T_{reg}$ cells in cluster 0 and cluster 8 shared this ST2 signature (Fig. 2c), only cells in cluster 8 showed high transcript expression of the immunosuppressive cytokine IL-10 (Fig. 2d). These $T_{reg}$ cell clusters (2 and 8) with highly suppressive properties were depleted in PI3Kδi-treated mice, whereas the clonally expanded cluster 5 $T_{reg}$ cells were enriched in PI3Kδi-treated mice showed a lack of transcripts associated with suppression (Fig. 2e, Extended Data Fig. 5c) and higher expression of several interferon-related response genes[21,22] (*Stat1*, *Stat3* and *Ifrd1*), suggestive of a pro-inflammatory environment (Extended Data Fig. 5a). Accordingly, $ST2^{+}Il10^{+}$ $T_{reg}$ cells were substantially reduced in PI3Kδi-treated mice (Fig. 2e). Notably, RNA velocity analysis, a tool to assess the developmental stage of cells in scRNA-seq data[23,24], infers a developmental trajectory over several progenitor states (cluster 2, 4 and then 0) culminating in clonally expanded ST2 $T_{reg}$ cells (cluster 8) in placebo-treated mice (Fig. 2f). These data indicate that PI3Kδ inhibition prevents the cellular differentiation into ST2 $T_{reg}$ cells, and instead diverts development to cluster 5 $T_{reg}$ cells that lack expression of transcripts associated with suppressive capacity, pointing to a possible mechanism for the onset of inflammation and colitis. We also observed a significant increase in CD8[+] T cells in colonic but not splenic tissue (Extended Data Fig. 5d, e). Colonic CD8[+] T cells expressed higher levels of PD-1 and ICOS upon PI3Kδ inhibition (Extended Data Fig. 5f, g), implying treatment-related changes in cell activation. Together, these findings suggest a heightened sensitivity of certain colonic $T_{reg}$ subsets

to PI3Kδi, potentially related to the high incidence of colitis observed in patients treated with PI3Kδi.

## PI3Kδ inhibition exacerbates colitis

To explore the connection between PI3Kδ inhibition and gastrointestinal toxicity in more detail, we used a dextran sulfate sodium-induced acute colitis model. Crucially, when compared with placebo-treated mice, we found that mice treated with PI3Kδi showed an accelerated and exacerbated disease phenotype, with a swift reduction in body weight and a higher overall colitis score characterized by significantly higher inflammation, crypt damage and area of infiltration (extent) (Fig. 3a, b), which indicate treatment-mediated alterations in tissue homeostasis driving immunopathology. To circumvent the emergence of these irAEs, we hypothesized that a transient depletion of $T_{reg}$ cells might suffice to restrict the immunosuppressive milieu in the tumour and thus drive anti-tumour immunity without causing substantial toxicity in non-malignant organs. We tested this hypothesis by using distinct treatment regimens, on which mice would either be kept on PI3Kδi for the duration of the experiment (continuous dosing), be kept on PI3Kδi for 4 days followed by 3 days off drug (intermittent dosing) or be kept on PI3Kδi for 2 days followed by 5 days off drug (infrequent dosing) for a total of two treatment cycles (Fig. 3c). All treatment conditions led to a decrease in tumour growth, albeit not significantly for the infrequent dosing condition, suggesting that transient interruptions of the immunosuppressive TME drive anti-tumour immunity. Most importantly, only continuous dosing led to increased CD8[+] T cell infiltration

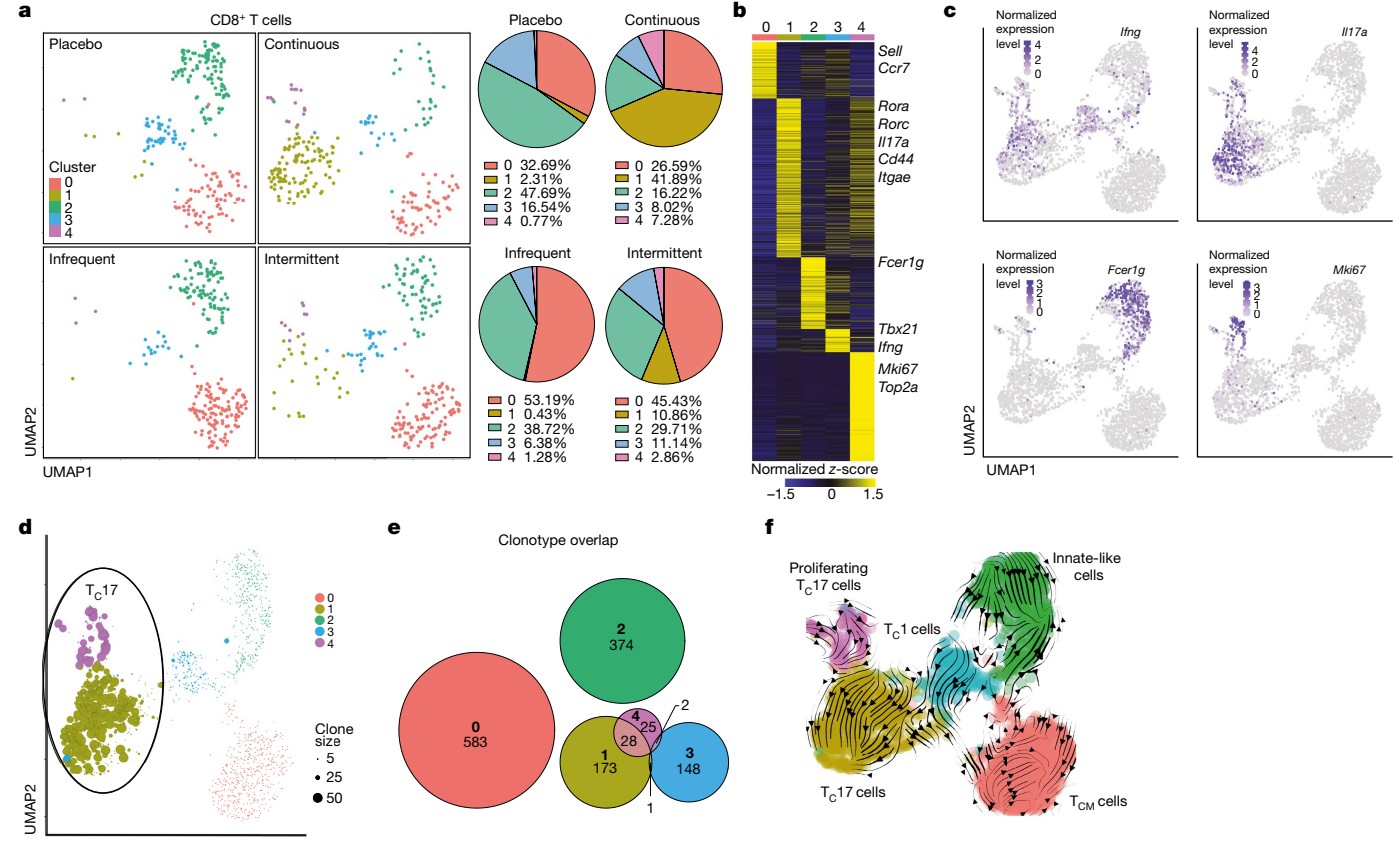

**Fig. 4 | Continuous dosing drives pathogenic T$_C$17 responses.** Mice were inoculated subcutaneously with B16F10-OVA cells and fed either a control diet or a diet containing PI-3065 inhibitor, with treatment conditions as in (Fig. 3c, d). **a**, Seurat clustering visualized by UMAP of CD8$^+$ T cells in colonic tissue at day 18 after tumour inoculation of mice treated as indicated. Pie charts depict the percentage of each cluster under the different treatment conditions. **b**, Heat map comparing gene expression of cells in all clusters. Depicted are transcripts that change in expression by more than 0.5-fold with adjusted

*P* values of ≤0.05. DEGs were called by MAST analysis; adjusted *P* values were calculated with the Benjamini–Hochberg method. *Sell* is also known as *Cd62l*; *Itgae* is also known as *Cd103*. **c**, Seurat-normalized expression of indicated genes in the different clusters. **d**, Clone size of cells in indicated clusters in UMAP space. **e**, Euler diagrams show the clonal overlap between CD8$^+$ T cells in the different clusters. **f**, RNA velocity analysis visualized by UMAP depicting likely developmental trajectories of CD8$^+$ T cells. Arrows indicate velocity streamlines. T$_C$ cells, cytotoxic T cells.

and decreased T$_{reg}$ cell levels in colonic tissue (Fig. 3d), indicating that intermittent dosing regimens might also decrease irAEs in human.

## Intermittent dosing curbs toxicity

To discern whether specific T cell subsets drive immunopathology on PI3Kδ inhibition, we performed scRNA-seq of colonic CD8$^+$ and CD4$^+$ T cells in the different treatment regimens. Unbiased clustering depicted by UMAP revealed five distinct CD8$^+$ cell clusters and six distinct CD4$^+$ T cell clusters (Fig. 4a, Extended Data Fig. 6a). In both instances, we identified a central memory T (T$_{CM}$) cell subset (cluster 0, red) expressing high levels of *Ccr7* and *Cd62L*, a T$_C$1 and T$_H$1 subset expressing high levels of interferon-γ transcripts (cluster 3, blue for T$_C$1 and cluster 1, ocher for T$_H$1), a T$_C$17 and T$_H$17 subset enriched for Il-17 transcripts (cluster 1, ochre for T$_C$17 and cluster 2, green for T$_H$17), and a proliferative subset that exhibited features of T$_C$17 (violet) or T$_H$17 (blue) cells, respectively (Fig. 4b, c, Extended Data Fig. 6b, c). Notably, we found dosing-dependent enrichment of the T$_C$17 and T$_H$17 subsets and pertaining proliferating clusters, making up approximately 50% of all cells in the continuous dosing regimen, whereas they were nearly completely absent in the other treatment conditions (Fig. 4a, Extended Data Fig. 6a). Of note, IL-17 producing cells have been shown to cause colitis[25–27]. Moreover, cells in these Il-17$^+$ clusters were heavily clonally expanded and exhibited substantial cellular and clonotypic overlap in both CD8$^+$ and CD4$^+$ T cells (Fig. 4d, e, Extended Data Fig. 6d, e),

probably contributing to their rapid expansion. Conversely, we found a dosing-dependent decrease of innate-like CD8$^+$ T cells, which have been implicated in controlling inflammation and the onset of colitis[28,29] (Fig. 4a–c). Last, RNA velocity analyses imply that the pathogenic T$_C$17 and T$_H$17 subsets are derived from IFN-γ-expressing progenitor cells (Fig. 4c, f, Extended Data Fig. 6c, f). Accordingly, T$_C$17 and T$_H$17 cells maintained high transcript expression of *Ifng* (Fig. 4c, Extended Data Fig. 6c).

Given that IL-10$^+$ ST2 T$_{reg}$ cells have been implicated in controlling IL-17 responses that would otherwise cause colitis[30], our data provide an explanation for the ripple effects ensuing after PI3Kδ inhibition that eventually cause irAEs. Specifically, our data imply that *Il10*-expressing ST2 T$_{reg}$ cells are highly susceptible to PI3Kδ inhibition, leading to a decrease in their abundance and thus to a disruption of gut homeostasis by causing a rapid expansion of pathogenic T$_H$17 and T$_C$17 cells that, together with a decrease in innate-like CD8$^+$ T cells, cause colitis. Moreover, intermittent PI3Kδi dosing provides the means to uncouple the anti-tumour effects from irAEs, providing ample rationale to test this concept in a follow-up clinical trial.

## Discussion

Here we find that in human and mouse tumour tissue, PI3Kδ inhibition leads to substantial changes in the cell composition of the TME by reducing the number of T$_{reg}$ cells and activating intratumoural CD4$^+$ and

CD8[+] T cells, which clonally expand and display heightened cytotoxic and cytolytic features. Notably, in mouse models, we find substantial changes in the transcriptional features and composition of colonic $T_{reg}$ cell subsets, which indicate that PI3Kδ inhibition affects $T_{reg}$ functionality, survival and tissue retention, thus altering $T_{reg}$ cell frequencies or $T_{reg}$ subtype compositions in both tumour and non-malignant tissues. These treatment-mediated changes, specifically the depletion of *Il10*-expressing ST2 $T_{reg}$ cells, is associated with colitis and expansion of pathogenic $T_C17$ and $T_H17$ T cell subsets in colonic tissue. Notably, these findings might be more broadly applicable, as tissue-resident ST2 $T_{reg}$ cells have been described in many non-malignant organs frequently affected by irAEs (for example, in skin) or might be affected by other $T_{reg}$ cell-targeting immunotherapies (for example, anti-CTLA-4). We show in mouse models that intermittent dosing with PI3Kδi is a rational treatment strategy that combines sustained anti-tumour immunity with reduced toxicity.

Our data show that the immunomodulatory effects of PI3Kδi need to be evaluated judiciously in treatment-naive patients unaffected by multiple lines of treatment and the immunosuppressive effect of haematological malignancies such as chronic lymphocytic leukaemia (CLL). It is clear that in the neoadjuvant setting in patients with HNSCC, at the evaluated doses and with daily scheduling, PI3Kδ inhibition has an unfavourable safety profile, limiting its feasibility and clinical benefit by causing frequent and severe grade 3/4 irAEs, probably driven by modulation of $T_{reg}$ cell behaviour in non-malignant tissues. On the basis of our findings, decreased dosages or an altered PI3Kδi treatment regimen will be required in solid tumours—especially in immune-competent patients—in order to be able to exploit the clear anti-tumour immune response induced by PI3Kδi while limiting the adverse effects associated with reduced $T_{reg}$ function in healthy tissues. Finally, our data suggest that the unique cellular composition of effector versus regulatory cells in the TME of each patient might be an important determinant of the efficacy of PI3Kδ inhibition. Thus, PI3Kδi might be especially useful in patients with high levels of intratumoural $T_{reg}$ cells and an unfavourable ratio of $T_{reg}$ versus CD8[+] tumour-infiltrating lymphocytes (TILs) in pre-treatment samples. Our study sets the stage for further exploration of PI3Kδ inhibitors as immunomodulatory agents in solid tumours.

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

# Methods

## Double-blind, randomized clinical trial and sample collection

To explore the immunomodulatory effects of PI3Kδ inhibition in humans, we conducted a multicenter, placebo-controlled phase II neoadjuvant trial with the PI3Kδi AMG319 in resectable HNSCC (Extended Data Fig. 1a, b; https://www.clinicaltrialsregister.eu/ctr-search/trial/2014-004388-20/results). All patients provided written informed consent for participation in the clinical trial. We focused on human papilloma virus (HPV)-negative HNSCC, as this cancer type is more prevalent, and because patients with this cancer type have poorer outcomes when compared to HPV-positive HNSCC, probably due to overall lower TIL infiltration[31–33]. The clinical trial was sponsored by Cancer Research UK Center for Drug Development (CRUKD/15/004) and approved by the Southampton and South West Hampshire Research Ethics Board; the trial EudraCT number is 2014-004388-20. Detailed information about the trial design, randomization procedure, protocol amendments, recruitment data, patient characteristics and adverse events have been deposited at https://www.clinicaltrialsregister.eu/ctr-search/trial/2014-004388-20/results#moreInformationSection and are in the CONSORT checklist. Patients were recruited after initial diagnosis and before definitive surgical treatment; drug treatment or placebo was given for up to 24 or 28 days respectively, prior to resection of tumour. In a previous phase I dose escalation study of heavily pretreated patients with either CLL or non-Hodgkin lymphoma, AMG319 doses of up to 400 mg were explored without reaching a maximally tolerated dose, and exhibited pharmacokinetic dynamics with a mean half-life of 3.8–6.6 h in plasma[34]. In that phase 1 study, daily dosing with 400 mg AMG319 led to near complete target inhibition (BCR-induced pAKT in ex vivo IgD-stimulated CLL samples) and >50% nodal regression[34], while irAE at grade 3 or above according to the common toxicity criteria (CTC) occurred after days 40 and 60. We thus reasoned that high grade irAEs were unlikely to occur during the shorter treatment duration in the neoadjuvant setting, and therefore selected 400 mg d$^{-1}$ as the starting dose. The intended time from initiating treatment with AMG319 or placebo to surgical resection of tumour was up to four weeks, with weekly blood draws. The full evaluation of radiological measurements has previously been reported at https://www.clinicaltrialsregister.eu/ctr-search/trial/2014-004388-20/results to the EU Clinical Trials Register in compliance with regulatory requirements. Primary endpoints were safety and assessment of CD8$^+$ immune infiltrates, secondary endpoints tumour responses and AMG319 pharmacokinetic evaluation (https://www.clinicaltrialsregister.eu/ctr-search/trial/2014-004388-20/results#endPointsSection). The sample size was calculated as follows: in a pilot cohort, the CD8 count in the biopsy taken at diagnosis, and in the resected tissue sample was quantified. The mean value at diagnosis was 25 cells per high power field (hpf), and this remained almost the same in the resected sample (26 cells per hpf). With an observed s.d. of 5 cells we posited we would observe a doubling to 50 cells per hpf following treatment with AMG 319, hence a difference between the two treatment groups of 25. To detect a standardized difference of 0.5 with 80% power and one-sided test of statistical significance of 20%, we required 36 patients to be randomized to AMG319 and 18 to placebo (54 in total). Randomization was at the level of the individual patient, using block randomization with randomly varying block sizes. During the course of the clinical trial the randomization list was held by the unblinded trial statistician and within the IWRS. Patients and care providers were blinded to the treatment allocation, and all immunological evaluations were completed by a pathologist and researchers who were blinded to the patient allocation to treatment arms. Patients were recruited from October 2015 to May 2018 in the UK (University Hospital Southampton NHS Foundation Trust, Poole Hospitals NHS Foundation Trust, Liverpool University Hospitals NHS Foundation Trust and Queen Elizabeth University Hospital Glasgow; two additional centres did not recruit patients); written informed consent was obtained from all subjects. Patients were eligible if they were ≥18 years of age, with histologically proven HNSCC for whom surgery was the primary treatment option, with laboratory results within specified ranges. Patients had to be clinically eligible for tumour resection; patients who had undergone prior radiotherapy, immunotherapy, chemotherapy or other anti-cancer therapy for their current HNSCC were excluded. Clinical data were obtained for age, gender, tumour size (T stage), and nodal status (N stage) (summarized in Source Data, Patient characteristics). Adverse event reporting was according to the National Cancer Institute CTCAE Version 4.02. Performance status and overall survival was collected to death or censored at last clinical review; clinical data were anonymized once the data had been collated and verified by the sponsor. Drug dosing was at 400 mg of the oral PI3Kδ inhibitor AMG319 (15 patients) and, after an independent safety review, dosing at 300 mg in 6 patients; all patients who had at least 4 doses of the drug were included in the final analyses. Radiological evaluation of change in tumour volume (Extended Data Fig. 1e) was undertaken by comparing baseline bi-dimensional measurements of tumour at baseline and before surgery. For response assessment, RECIST 1.1 was used. The full data on radiological measurements is available at https://www.clinicaltrialsregister.eu/ctr-search/trial/2014-004388-20/results in the EU Clinical Trials Register in compliance with regulatory requirements. The study was discontinued after 30 (of the target sample size of 54) patients had been dosed with AMG319 or placebo, thus limiting the clinical information on outcomes that can be gained from this trial. All patients had tissue collected as a dedicated research biopsy after consent and prior to randomization, with an additional sample collected during surgical resection. Tumour tissue was obtained fresh on the day of biopsy or surgery and a sample was immediately snap frozen. A proportion of the tumour tissue was cryopreserved in freezing medium (90% FBS and 10% DMSO) for subsequent analyses or, alternatively, directly disaggregated using a combination of enzymatic and mechanical dissociation for immediate analysis by FACS or cryopreservation as a single-cell suspension, as previously described[35]. Blood samples were collected during the course of the study from which plasma and peripheral blood mononuclear cells (PBMCs) were collected. PBMCs were isolated by centrifugation over lymphoprep (Axis-Shield PoC AS).

## Histology and immunohistochemistry

Double immunostaining for CD8 and FOXP3 was performed on a Leica Bond RX platform, with antigen retrieval performed for 20 min at 97 °C Bond ER2 antigen retrieval solution. Primary antibodies were incubated for 30 min at room temperature (FoxP3 - Abcam: Clone 236A/E7 1:100 dilution; CD8 - DAKO: Clone C8/144B 1:50 dilution) and detected using the Leica Refine Polymer brown and red detection systems. Analysis was performed by two independent and blinded head and neck pathologists counting intratumoural CD8$^+$ and FOXP3$^+$ TIL in multiple random high-power fields at 200× magnification. Where possible, ten high-power fields were counted.

## Pharmacokinetics of AMG319

Fifty microlitres of thawed plasma samples were mixed with 300 μl of extraction solution (100 ng ml$^{-1}$ [2H$^3$, N$^{15}$]-AMG319 in methanol), centrifuged at 10,000$g$ for 5 min to precipitate the plasma proteins. The supernatant was transferred to a UPLC vial and placed on the autosampler (maintained at 8 °C) for analysis. A freshly prepared calibration curve in the range 1–1,000 ng ml$^{-1}$ and frozen QC samples at 10, 100, 500 and 1,000 ng ml$^{-1}$ (K2 EDTA human plasma spiked with AMG319) were analysed alongside each batch of patient samples. Five microlitres of supernatant was injected into the UPLC-MS/MS system, configured with a Waters Acquity UPLC and Waters Quattro Premier XE mass spectrometer. Analytes were separated on an Acquity UPLC BEH C18 1.7 μm (2.1 mm × 100 mm) column with a mobile phase flow rate of 0.3 ml min$^{-1}$. Mobile phase was composed of water, acetonitrile

and formic acid. Analytes were detected using the multiple reaction monitoring (MRM) mode of the MS/MS system, operating in positive ion electrospray mode. MRMs were set up at $m/z$ 386.4→251.3, 386.4→236.6, 251.3→251.3 and 251.3→236.3 for AMG319 and at $m/z$ 390.5→254.4 for [2H$^3$, N$^{15}$]-AMG319. MassLynx software (version 4.1, Waters Ltd.) was used to control the instrumentation and for analysis of the peaks of interest and processing of spectral data.

## pAKT measurement

Whole blood samples (10 ml) were collected in sodium heparin tubes pre-dose and 4 h post dose on days 1 and 15 for the first 11 patients (day 8 and 15 for the remaining 19 patients). Blood was stimulated with double-diluted anti-IgD (25–0.008 µg ml$^{-1}$) in deep well plates for 5 min. Blood was then lysed and fixed with BD PhosFlow Lyse/Fix buffer. Cell pellets were washed and then stored at −80 °C until all samples from the same patient were ready for further analysis. Upon thawing, cells pellets were incubated with anti-human CD3-FITC and CD14-FITC, washed in PBS + 1% FBS, permeabilized with 80% MeOH and washed again before intracellular staining with CD20-PE Cy7 and pAKT (S473). Stained cell pellets were washed again before staining with a secondary antibody (anti-rabbit Alexa 647). Events were subsequently acquired on a Canto II flow cytometer (BD Bioscience), and analysed using FACS Diva. MFI of pAKT in B cells was plotted against the anti-IgD concentration, which was used to activate the B cells. The area under the curve was calculated and a drop of 50% in area under the curve between pre- and post-dose was validated to be the result of drug inhibition.

## Mice

C57BL/6J (JAX stock no. 000664), OT-I (JAX stock no. 003831), *Rag1*$^{-/-}$ (JAX stock no. 002216) and *CD8*$^{-/-}$ (JAX stock no. 002665) mice were obtained from Jackson labs. *Foxp3*$^{RFP}$ mice (JAX stock no. 008374) were a gift from K. Ley. Age (6–12 weeks) and sex-matched mice were used for all experiments. The housing temperature was controlled, ranging from 20.5–24 °C, humidity was monitored but not controlled and ranges from 30–70%. The 12 h daily light cycle was from 06:00 to 18:00. All animal work was approved by the relevant La Jolla institute for Immunology Institutional Animal Care and Use Committee.

## Tumour experiments

Mice were inoculated with $1 \times 10^5$ to $1.5 \times 10^5$ B16F10-OVA cells subcutaneously into the right flank. Mice were put on either a control diet or a diet containing the PI3Kδi PI-3065 on day 1 or day 5 after tumour inoculation. Diets were prepared using powdered 2018 global rodent diet (Envigo) mixed with or without PI-3065 at 0.5 g kg$^{-1}$, which corresponds to a daily dose of 75 mg kg$^{-1}$ as used in our previous study[8]. To pellet the food, 50% v/w water was added to the diet and dough thoroughly mixed, compressed, moulded and dried before use. Tumour size was monitored every other day, and tumour harvested at indicated time points for analysis of tumour-infiltrating lymphocytes. Tumour size limit of 15 mm in diameter was not exceeded and volume was calculated as $\frac{1}{2} \times D \times d^2$, where $D$ is the major axis and $d$ is the minor axis, as described[36].

## Bulk transcriptome analyses

Cryosections (10 µm) were cut from snap frozen tumour and RNA was extracted using the Maxwell RSC instrument and Maxwell RSC SimplyRNA Tissue kit (Promega), according to the manufacturer's instructions. RNA was quantified using the Qubit fluorometer (ThermoFisher Scientific) and quality was assessed using the Agilent 2100 Bioanalyzer generating an RNA integrity number (RIN; Agilent Technologies). RNA sequencing was performed by Edinburgh Genomics; mRNA libraries were prepared using the TruSeq Stranded Total RNA Library Prep Kit (Illumina) and paired-end sequenced (100 bp) on the NovaSeq 6000 platform (Illumina) to yield an average read depth of $40 \times 10^6$ reads. Reads were mapped to hg19 reference genome using STAR with our in-house pipeline (https://github.com/ndu-UCSD/

LJI_RNA_SEQ_PIPELINE_V2). A total of 22 paired (14 from treatment and 8 from placebo group) samples with at least 70% of mapping reads were selected. Differential expression analysis between the pre and post treatment, as well as between pre and post placebo, was performed using DESeq2 (v1.24.0). The threshold for DEGs was determined with fold change of >log$_2$ 0.75 and an adjusted $P$ value <0.1. Between treatment pre and post, 93 genes were identified as significant, whereas 3 genes were significant between placebo pre and post. Cells were dispersed from fresh tumour tissue and used immediately for flow cytometric analysis and cell sorting. CD8$^+$ T cells were bulk sorted into ice-cold TRIzol LS reagent[35] (Thermo Fisher Scientific) on a BD FACS Fusion (BD Bioscience). Reads from sorted CD8 RNA were mapped to hg19 reference genome using STAR with the same in-house pipeline as above. In total, we had 17 samples available, placebo (2 pre-treatment and 3 post-treatment) and treatment (6 pre-treatment and 6 post-treatment), out of which 3 were paired (1 placebo and 2 treatment). The DEGs between post treatment and remaining samples resulted in 455 significant genes (fold change of >log$_2$ 0.75 and an adjusted $P$ value <0.05).

## Flow cytometry

Cells dispersed from cryopreserved tumour tissue or PBMCs were prepared in staining buffer (PBS with 2% FBS and 2 mM EDTA), FcR blocked (clone 2.4G2, BD Biosciences) and stained with antibodies as indicated below for 30 min at 4 °C. Cell viability was determined using fixable viability dye (ThermoFisher).

Mouse lymphocytes were isolated from the spleen by mechanical dispersion through a 70-µm cell strainer (Miltenyi) to generate single-cell suspensions. RBC lysis (Biolegend) was performed to remove red blood cells. Tumour samples were harvested and lymphocytes were isolated by dispersing the tumour tissue in 2 ml of PBS, followed by incubation of samples at 37 °C for 15 min with DNase I (Sigma) and Liberase DL (Roche). The suspension was then diluted with MACS buffer and passed through a 70-µm cell strainer to generate a single-cell suspension. Colons were collected and rinsed in 1 mM dithiothreitol to remove faeces. Each colon was cut into 2–3 mm pieces and incubated 3 times in pre-digestion solution (HBSS containing 5% FBS and 2 mM EDTA) at 37 °C for 20 min under high rotation to remove epithelial cells. Then tissues were minced with scissors and incubated with digestion solution (HBSS containing 5% FBS, 100 µg ml$^{-1}$ DNase I (Sigma) and 1 mg ml$^{-1}$ collagenase (Sigma)) at 37 °C for 20 min under high rotation to get single-cell solutions of lamina propria cells. Cells were prepared in staining buffer (PBS with 2% FBS and 2 mM EDTA), FcR blocked (clone 2.4G2, BD Biosciences) and stained with antibodies as indicated below for 30 min at 4 °C; secondary stains were done for selected markers. Samples were then sorted or fixed and intracellularly stained using a FoxP3 transcription factor kit according to manufacturer's instructions (eBioscience). Cell viability was determined using fixable viability dye (ThermoFisher). The following antibodies from BD Biosciences, Biolegend, Miltenyi or eBbioscience were used: anti-human PD-1 (EH12.1, 1:30), CD4 (OKT4, 1:30), CD137 (4B4-1, 1:30), GITR (108-17, 1:30), ICOS (C398.4A, 1:50), CD8A (SK1, 1:30), CD25 (M-A251, 1:20), CD3 (SK7, 1:30), CD127 (eBioRDR5, 1:50), CD45 (HI30, 1:30), CD14 (HCD14, 1:50), CD20 (2H7, 1:50); anti-mouse CD3 (145-2C11, 1:100), CD4 (RM4-5, 1:100), CD8 (53-6.7, 1:100), PD-1 (29F1.A12, 1:100), ST2 (U29-93, 1:100) Ki67 (B56, 1:40), TOX (REA473, 1:40), CD19 (6D5, 1:100), CD45 (30-F11, 1:100), FOXP3 (FJK-16s, 1:100) and GZMB (QA16A02, 1:40). All samples were acquired on a BD FACS Fortessa or sorted on a BD FACS Fusion (both BD Biosciences) and analysed using FlowJo 10.4.1 for subsequent scRNA-seq.

## Colitis experiments

DSS (molecular mass ≈ 40,000) (Alfa Aesar) 2.5% (w/v) was added to the drinking water of mice with ad libitum access. Body weight of the mice was monitored. Colon tissues were collected for histological

analysis at the end point. Whole colons were harvested from mice between cecum and rectum. Stools were flushed out of lumen with PBS. Then colons were fixed with zinc formalin for 5 min. Fixed colons were opened longitudinally, flattened, cut into 3 fragments and further fixed in zinc formalin for 48 h in cassettes. After fixation, samples were transferred to 70% isopropanol for long term storage or H&E staining. Slides were scored blindly according to the following criteria: inflammation, area of infiltration (extent), crypt damage and oedema. The colon was divided into three equal parts and the middle section was utilized for scoring according to system shown in Extended Data Table 1. Four randomly selected areas were analysed and a histological score was determined.

### Single-cell transcriptome analysis

**Human.** scRNA-seq was performed by Smart-seq2 as described[37]. Reads were mapped with our in-house pipeline as above. Good quality cells were defined as those with at least 200 genes, at least 60 percent of mapping reads, mitochondrial counts of at most 20%, at least 50,000 total counts (reported by STAR excluding tRNA and rRNA), and a 5′ to 3′ bias of at most 2. Filtered cells were analysed using the package Seurat (v3.1.5). In order to separate CD4 and CD8 more effectively, we performed differential gene expression analysis between single-positive cells using *CD4* and *CD8B* genes. Cells were clustered using 178 significant genes (adjusted *P* value < 0.05).

**Mouse T$_{reg}$ cells.** scRNA-seq was performed using the 10x platform (10x Genomics) according to the manufacturer's instructions. Reads were mapped with Cell Ranger followed by our in-house QC pipeline (https://github.com/vijaybioinfo/quality_control) and demultiplexed with bcl2fastq using default parameters. The Cell Ranger aggr routine was used and CITE-seq data was processed using our custom pipeline (https://github.com/vijaybioinfo/ab_capture). In brief, raw output from Cell Ranger was taken and cell barcodes with less than 100 unique molecular identifier (UMI) counts as their top feature were discarded and the remaining barcodes were classified by MULTIseqDemux from Seurat. Finally, cell barcodes where the assigned feature did not have the highest UMI count were fixed, and cells with a fold change of less than 3 between the top two features were reclassified as doublets. Before clustering, cells were filtered for at least 300 and at most 5,000 genes, at least 500 and at most 10,000 UMI counts, and at most 5% of mitochondrial counts. Cell types were identified using Seurat's FindAllMarkers function. Differential expression was calculated with MAST[38] (v1.10.0) DESeq2 (v1.24.0) as previously described[37] and genes with an adjusted *P* value < 0.05 and a fold change of >log$_2$ 0.5 were defined as significant. P-values were corrected for multiple comparisons using the Benjamini–Hochberg method. Gene Set Enrichment Analysis (GSEA) scores were estimated with fgsea (v1.10.1) in R using signal-to-noise ratio as the metric (minSize = 3 and maxSize = 500). Enrichment scores were shown as GSEA plots. Signature scores were computed using Seurat's AddModuleScore function with default parameters. In short, the score is defined for each cell by subtracting the mean expression of an aggregate of control gene lists from the mean of the signature gene list. Control gene lists were randomly selected (same size as the signature list) from bins delimited based on the level of expression of the signature list.

**Mouse colonic CD4$^+$ and CD8$^+$ T cells.** scRNA-seq was performed by using the 10x platform. Mapping, aggregation, and QC was carried out as described above with the following thresholds: genes per cells range of [300; 4,500], UMI content per cell was [500; 20,000], percent of mitochondrial counts of ≤10%, and a doublet score of ≤0.3. Clusters of contaminant cells expressing epithelial, monocyte, and fibroblast markers were eliminated after the first round of clustering. The final number of cells comprised $n$ = 6,415 CD4$^+$ T cells and $n$ = 2,715 CD8$^+$ T cells.

### T cell receptor analysis

TCRs were reconstructed from scRNA-seq reads using MiXCR with default parameters. Then, shared TCRs were defined by having the same CDR3 sequence in both the alpha and beta chains and coming from the same donor. Enriched TCR were defined as those with a frequency higher or equal to two. Finally, TCR network plots were generated using the Python package graphviz.

### Quantification and statistical analysis

The number of subjects, samples or mice per group, replication in independent experiments and statistical tests can be found in the figure legends. Details on quality control, sample elimination and displayed data are stated in the methods and figure legends. Sample sizes were chosen based on published studies to ensure sufficient numbers of mice in each group enabling reliable statistical testing and accounting for variability. RNA-seq samples that did not pass the QC check were excluded from downstream analyses. Experimental results were reliably reproduced in at least two independent experiments. Animals of same sex and age were randomly assigned to experimental groups, and blinding was not performed. Extended Data Figs. 1a, 3a were created with BioRender.com, the statistical analyses were performed with Graph Pad Prism 9 and statistical tests used are indicated in the figure legends.

### Reporting summary

Further information on research design is available in the Nature Research Reporting Summary linked to this paper.

## Data availability

Sequencing data has been uploaded onto the Gene Expression Omnibus (accession code GSE166150). Source data are provided with this paper.

## Code availability

Scripts used for this paper and explanations thereof are available at our GitHub repository (https://github.com/vijaybioinfo/PI3Kd_2022).

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

**Acknowledgements** We thank C. Rommel, K. Ali, P. Johnson and D. Scott for support and help during the initial phases of setting up the CRUK/Amgen collaboration; F. Martin for project support over the years; M. Lopez and the research pathology core at University Hospital Southampton for performing the immunohistochemistry on the paraffin embedded material; I. Henderson for help with the tetanus ELISAs; the La Jolla Institute (LJI) Flow Cytometry Core for assisting with cell sorting; the LJI sequencing core for the bulk and single-cell sequencing; the LJI histology core for processing the mouse histology samples; G. Means for method and training for the phospho-AKT assays; P. Friedmann for constructive critique of the paper; and D. Singh for advice on and help with experimental work. This clinical trial work was funded by a CDD trial Grant CRUKD/15/004 (C.H.O.), a Cancer Research UK Centres Network Accelerator Award Grant (A21998) (O.W., K.M. and J.T.), the CRUK and NIHR Experimental Cancer Medicine Center (ECMC) Southampton (A15581), the CRUK and NIHR ECMC Liverpool (A25153), Cancer Research UK Programme Grant (C23338/A25722 (E.L.-G. and B.V.)); the UK NIHR UCLH Biomedical Research Centre (B.V.), S10OD025052 (Illumina Novaseq6000), S10RR027366

(FACSAria II cell sorter), NIH grant P01 DK46763 (M.K.), the William K. Bowes Jr Foundation (P.V.), Whittaker iCure Foundation (P.V., L.C. and C.H.O.), the Deutsche Forschungsgemeinschaft DFG research fellowship no. WI 5255/1-1:1 (A.v.W.) and Erwin Schrödinger Fellowship (M.D.). The clinical delivery of this work was supported by the Wessex Clinical Research Network and National Institute of Health Research UK. We further acknowledge Cancer Research UK (Centre for Drug Development) as the clinical trial Sponsor and for funding and management of the Phase II clinical trial, as well as Amgen for supply of the PI3Kδi AMG319.

**Author contributions** S.E.: study design, experimental work, data interpretation and writing. C.R.-S.: bioinformatic evaluation, data interpretation and paper review. Y.L.: experimental work and paper review. E.K.: study design, patient recruitment and paper review. L.C.: data collation, data interpretation, paper writing and review. J.T.: bioinformatic evaluation, data interpretation and paper review. O.W.: experimental work, data interpretation and paper review. A.v.W.: experimental work, data interpretation and paper review. D.J.: experimental work, data interpretation and paper review. K.M.: experimental work, data interpretation and paper review. H.S., M.M. and A.W.: experimental work and paper review. E.L.-G.: experimental work, provision of study materials, data interpretation and paper review. T.-F.C.: experimental work. M.D.: experimental work. N.A.D., L.E. and F.K.: study conduct, safety data review and monitoring, data review and verification for sponsor. G.A.: study design, safety review and data review. G.H.: generation and provision of placebo and IMP. J.J.S., A.G.S., R.S., J.A.M., C.P., J.H.D., P.A.B. and R.P.S.: patient recruitment and paper review. P.L.: data generation and interpretation and paper review. W.W.: study design and statistical review for sponsor. A.H.: study design and statistical review. G.J.T.: histopathological evaluation, data generation and interpretation and paper review. T.M.J.: study development, patient recruitment and paper review. F.A.: bioinformatic analyses and supervision, data review and paper review. G.S.: RNA sequencing and quality control supervision. K.O.: study design and paper writing and review. G.F.: study development. M.K.: study design, paper writing and review. B.V.: study design, paper writing and review. P.V.: study design, data generation and review, paper writing and review. C.H.O.: study design, patient recruitment, data generation and review, paper writing and review. P.V. and C.H.O. conceived, supervised and led the work.

**Competing interests** G.F. is an employee of Amgen Inc. B.V. is a consultant for iOnctura (Geneva, Switzerland), Venthera (Palo Alto, US) and Olema Pharmaceuticals (San Francisco, US) and has received speaker fees from Gilead (Foster City, US). K.O. has received consultancy fees from iOnctura, Macomics, Gilead Sciences and Karus Therapeutics and has received research funding from GSK. M.K. is on the scientific advisory board of Prometheus. C.H.O. led the clinical trial of AMG319 with funding by Cancer Research UK, Amgen provided clinical grade compound free of charge for this trial. All other authors declare no conflicts of interest.

**Additional information**
**Correspondence and requests for materials** should be addressed to Pandurangan Vijayanand or Christian H. Ottensmeier.

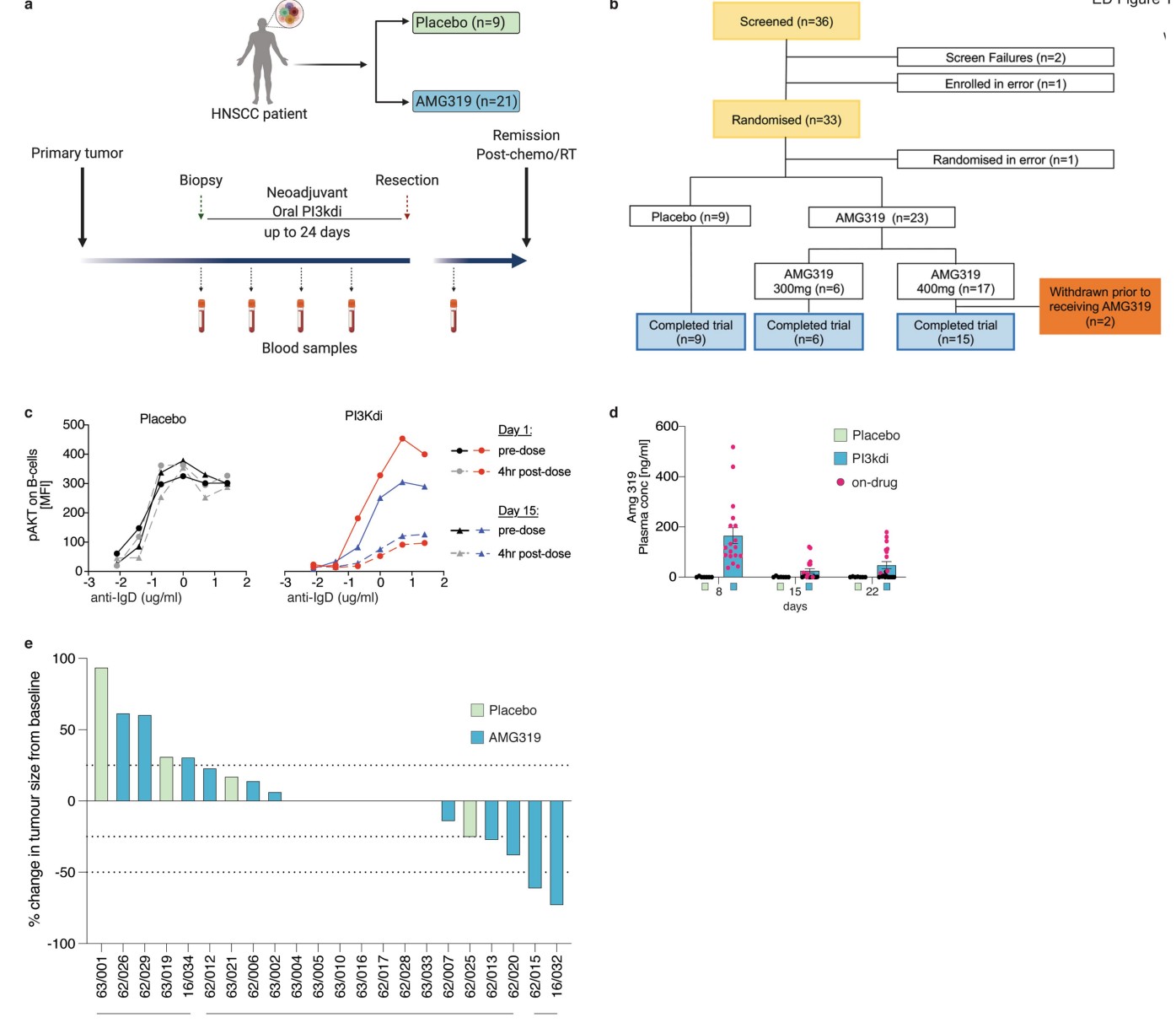

**Extended Data Fig. 1 | Trial schematic, pharmacokinetic & pharmacodynamic assessments and tumor response evaluation. a**, Trial schematic of the placebo-controlled randomized phase II study. **b**, Consort workflow, 36 patients were screened, of which 33 were recruited and randomly allocated to the placebo control arm or AMG319 drug-treatment arm; 30 patients ultimately received at least one dose of either AMG319 or placebo. Of the 21 patients that were treated with AMG319, 6 patients received daily doses of 300 mg and 15 patients (2 patients withdrew consent prior to receiving the first dose) received daily doses of 400 mg. An initial biopsy was taken before trial initiation and surgical resection of tumors was performed 4–6 weeks after the first dose of treatment. Pre- on- and post-treatment blood samples were collected for further analysis. **c**, **d**, Assessment of the level of AKT phosphorylation in B cells at indicated time points pre-dose and 4h after treatment with AMG319, data from one representative patient are shown. **d**, Plasma concentrations of AMG319 in placebo-controlled and drug-treated patients at indicated time points, n = 9 patients for the Placebo group and n = 18 patients for the AMG319 group. Highlighted in red are patients who were either on-treatment or had only recently (2 days prior to analysis) or briefly discontinued treatment. **e**, Waterfall plot depicting the change in tumor volume from screening to pre-surgery measured by MRI scan in patients treated with AMG319 (blue bars) or placebo (green bars), shown are all patients for which MRI scans have been performed. Data in (**d**) are mean +/− S.E.M.

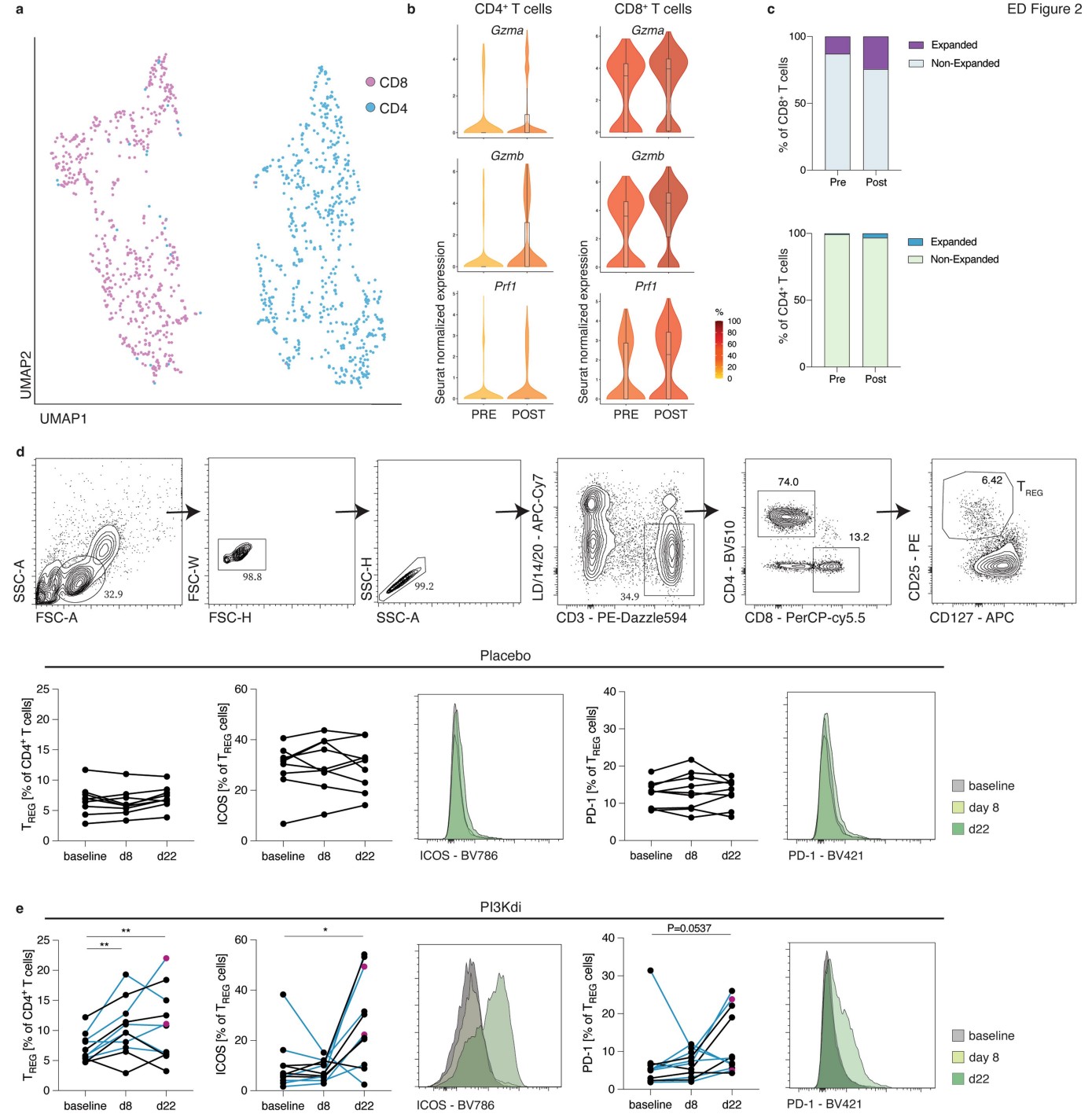

ED Figure 2

**Extended Data Fig. 2 | Single-cell RNA-seq analysis reveals substantial oligoclonal expansion of tumor-infiltrating CD8+ T cells post-treatment.** **a**, Analysis of smart-seq2 single-cell RNA-seq data of sorted tumor-infiltrating CD3+ T cells from patients 20, 30, 32, 33, 34 and 35 displayed by UMAP analysis. **b**, Violin plots depicting the Seurat normalized expression of differentially expressed highlighted genes in CD4+ T cells (left) or CD8+ T cells (right) of the 6 patients pertaining to **a**, the center line depicts the median, edges delineate the 25th and 75th percentiles and whiskers depict minimum and maximum values. **c**, Percentages of non-expanded and expanded CD8+ and CD4+ T cell

clones in pre-*versus* post-treatment samples. **d**, **e**, flow-cytometric analyses of the frequency of and expression of activation markers in circulating $T_{REG}$ cells in placebo-treated (d) and AMG319-treated patients (**e**), P = 0.0098 for the frequency of circulating $T_{REG}$ cells (baseline vs d8 and baseline vs d22), P = 0.0234 for ICOS+ $T_{REG}$ cells, at indicated time points; the blue lines depict patients with grade 3/4 irAEs and the red dots indicate the patients with CR/PR. Data are mean +/− S.E.M. Significance for comparisons were computed using two-tailed Wilcoxon matched-pairs signed rank test between baseline and d8 or d22 respectively.

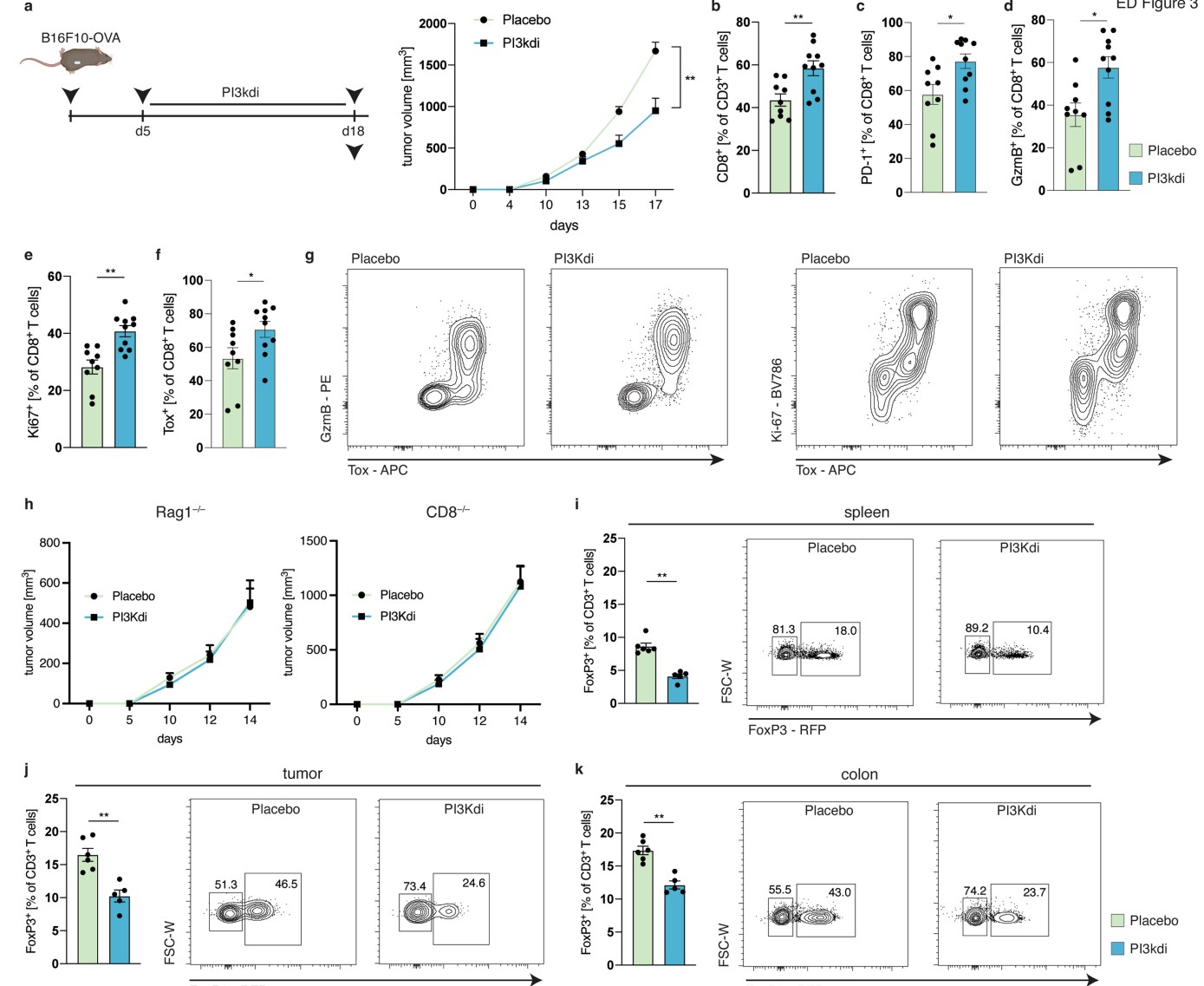

**Extended Data Fig. 3 | PI3Kδ-inhibition induces are pro-inflammatory tumor microenvironment.** Mice were inoculated *s.c.* with B16F10-OVA cells and fed either a control diet or a diet containing the PI-3065 PI3Kδ inhibitor for the indicated treatment period. Tumor volume (**a**) and flow-cytometric analyses of cell frequencies (**b**–**g**) of mice treated as indicated; P = 0.003 (a), P = 0.0076 (b), P = 0.0279 (c), P = 0.0172 (d), P = 0.0013 (e), P = 0.0435 (f), n = 9 mice for the Placebo group and n = 10 mice for the PI3Kδ group for a-f **g**, shown are representative contour plots of intratumoral CD8⁺ T cells depicting the indicated markers. **h**, Tumor volume of Rag1⁻/⁻ or CD8⁻/⁻ mice treated as indicated, n = 6 mice/group for Rag1⁻/⁻ and n = 5 mice/group for CD8⁻/⁻ mice.

**i–k**, flow-cytometric analyses of $T_{REG}$ cell frequencies (**b**) in indicated organs of mice treated as indicated, n = 6 mice for the Placebo group and n = 5 mice for the PI3Kδ group. Shown are representative contour plots of FoxP3-expressing (RFP⁺) CD4⁺ T cells in indicated organs; P = 0.0043 (spleen), P = 0.0043 (tumor), P = 0.0043 (colon). Not significant, P = 0.1234; *P = 0.0332; ***P = 0.0002; and ****P < 0.0001. Data are mean +/− S.E.M and statistical significance for comparisons was computed using two-tailed Mann-Whitney test; data are representative of at least two independent experiments. DEGs in (**b**–**e**) were called using MAST and adjusted p-values were calculated with the Benjamini-Hochberg method.

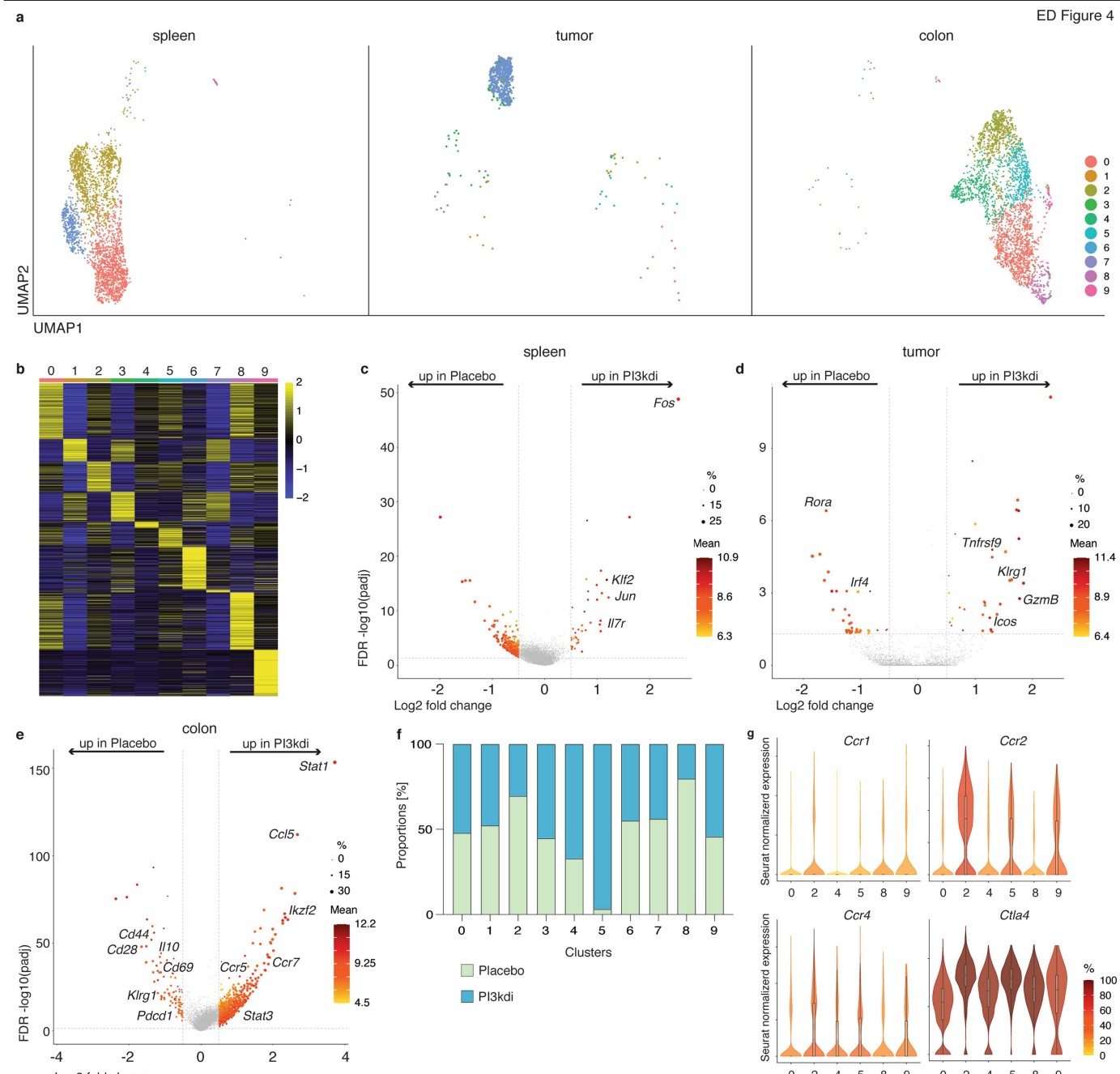

**Extended Data Fig. 4 | T$_{REG}$ cells in different tissues exhibit unique transcriptomic signatures. a**, Analysis of 10x single-cell RNA-seq data displayed by UMAP analysis. Seurat clustering of FoxP3$^+$CD4$^+$ T cells in spleen (left), tumor (middle) and colon (right). **b**, Heatmap comparing gene expression of cells in all clusters. Depicted are transcripts that change in expression more than 0.5-fold and adjusted $P$ value of ≤ 0.05. **c–e**, Volcano plots of single-cell RNA-seq analysis of placebo-treated control mice and PI-3065-treated mice in spleen (**c**) tumor (**d**) and colon (**e**). Highlighted are transcripts with a >0.5 log2 fold change. **f**, Bar charts depicting the proportion of cells in each cluster. Bars are colorized based on cells in indicated treatments making up the cluster. **g**, Violin plots showing normalized expression levels (log$_2$(CPM+1)) of highlighted genes in cluster the colonic clusters pertaining to Fig. 2a, b, the center line depicts the median, edges delineate the 25th and 75th percentiles and whiskers depict minimum and maximum values.

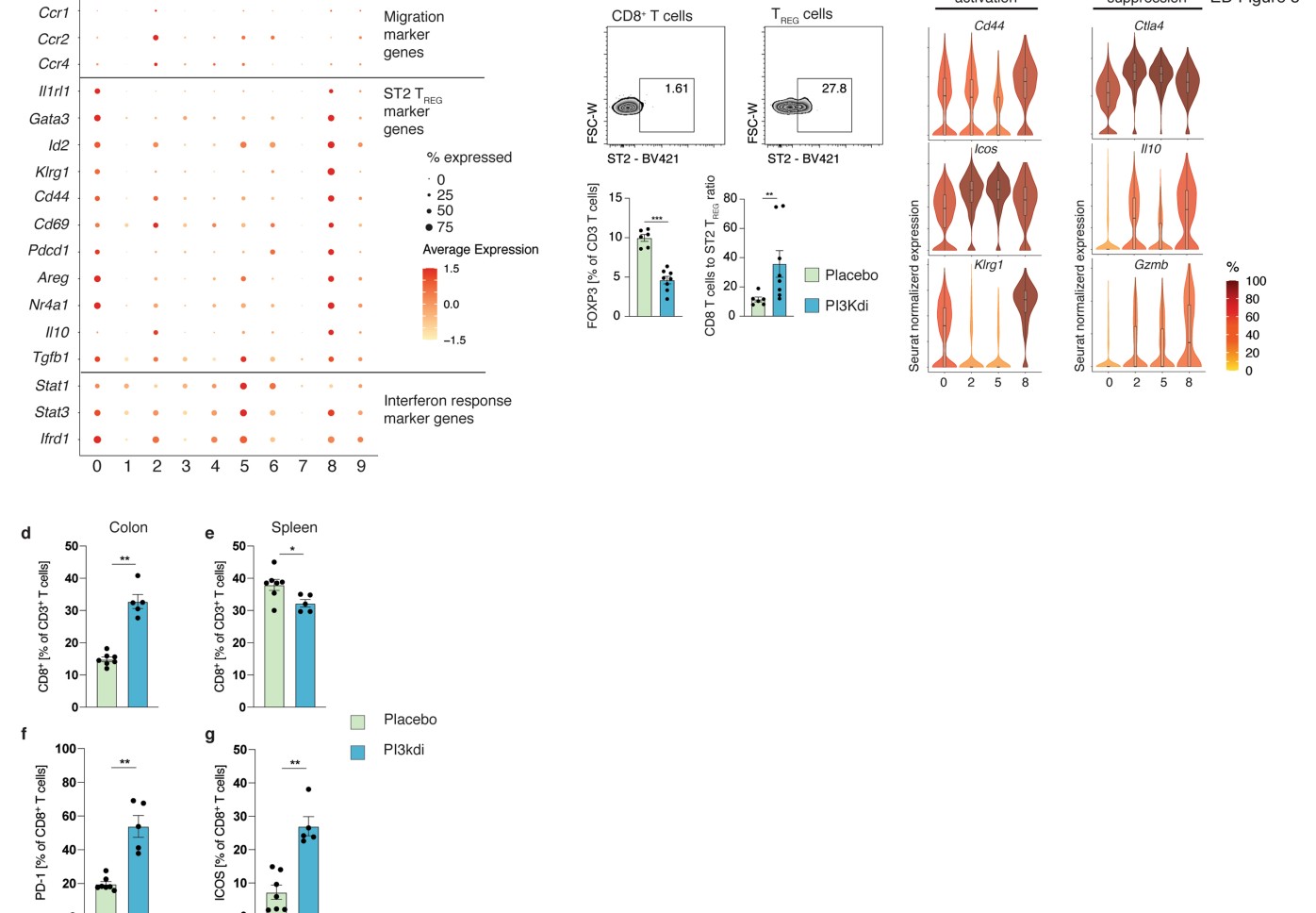

**Extended Data Fig. 5 | Colonic ST2 T$_{REG}$ cells exhibit features of superior suppressive capacity. a**, Curtain plot highlighting selected genes in each cluster with average transcript expression (color scale) and percent of expressing cells (size scale). **b**, Flow-cytometric analyses depicting the expression of ST2 in CD8$^+$ T cells or CD4$^+$ T$_{REG}$ cells in representative zebra plots (left), the frequency of ST2$^+$ T$_{REG}$ cells (n.s., P = 0.0549), FOXP3$^+$ cells (P = 0.0007), and the ratio of CD8$^+$ T cells to ST2 T$_{REG}$ cells (P = 0.0047) in placebo-treated (n = 6 mice) and PI3Kδi-treated mice (n = 8 mice) **c**, Violin plots showing normalized expression levels (log$_2$(CPM+1)) of highlighted genes in indicated clusters pertaining to Fig. 2a, b, the center line depicts the median,

edges delineate the 25th and 75th percentiles and whiskers depict minimum and maximum values. **d–g**, Flow-cytometric analyses of CD8$^+$ T cell frequencies in colon (**d**) P = 0.0025, spleen (**e**) P = 0.0013 and of the expression of PD-1 (**f**) P = 0.0025 and ICOS (**g**) P = 0.0025 on colonic CD8$^+$ T cells, n = 7 mice for the Placebo group and n = 5 mice for the PI3Kδ group. Not significant, P = 0.1234; *P = 0.0332; ***P = 0.0002; and ****P < 0.0001. Data (**b, d–g**) are mean +/− S.E.M and statistical significance for comparisons was computed using two-tailed Mann-Whitney test; data are representative of at least 2 independent experiments.

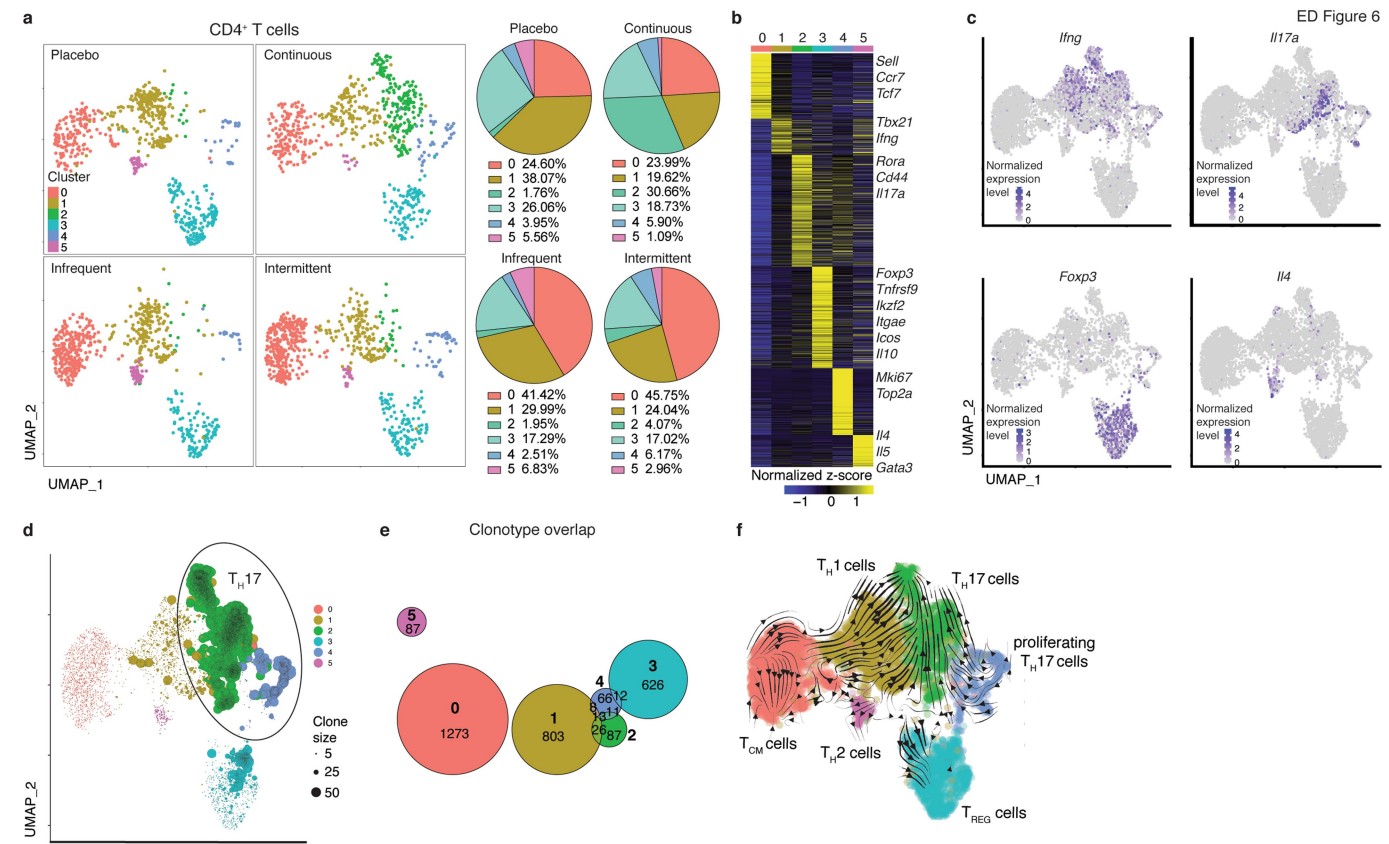

**Extended Data Fig. 6 | Continuous dosing drives pathogenic T$_H$17 responses.** Mice were inoculated *s.c.* with B16F10-OVA cells and fed either a control diet or a diet containing the PI-3065 PI3Kδ inhibitor, treatment conditions as in (Fig. 3c, d). **a**, depicted is Seurat clustering, visualized by UAMP, of CD4$^+$ T cells in colonic tissue at day 18 after tumor inoculation of mice treated as indicated, pie charts depict the percentage of each cluster in the different treatment conditions. **b**, Heatmap comparing gene expression of cells in all clusters. Depicted are transcripts that change in expression more

than 0.5-fold and adjusted *P* value of ≤ 0.05, DEGs were called by MAST analysis, adjusted p-values were calculated with the Benjamini-Hochberg method. **c**, Seurat-normalized expression of *Ifng* (top left), *Il17a* (top right), *Foxp3* (bottom left) and *Il4* (bottom right) in the different clusters. **d**, Clone size of cells in indicated clusters in UMAP space. **e**, Euler diagrams show the clonal overlap between the CD4$^+$ T cells in the different clusters. **f**, RNA velocity analysis visualized by UMAP depicting likely developmental trajectories of CD4$^+$ T cells, arrows indicate velocity streamlines.

**Extended Data Table 1 | Colitis scoring system**

| Inflammation | Area of inflammation | Crypt damage | Edema |
|---|---|---|---|
| 0 = none | 0 = none | 0 = none | 0 = none |
| 1 = mild | 1 = mucosa | 1 = basal 1/3 damaged | 1 = ≤ 2x submucosal thickness |
| 1 = moderate | 2 = mucosa & submucosa | 2 = basal 2/3 damaged | 2 = ≥ 2x submucosal thickness |
| 3 = severe | 3 = transmural | 3 = only surface epithelium intact | |
| | | 4 = entire crypt and epithelium lost | |

# nature research

|---|---|

# Reporting Summary

Nature Research wishes to improve the reproducibility of the work that we publish. This form provides structure for consistency and transparency in reporting. For further information on Nature Research policies, see Authors & Referees and the Editorial Policy Checklist.

## Statistics

For all statistical analyses, confirm that the following items are present in the figure legend, table legend, main text, or Methods section.

| n/a | Confirmed | |
|---|---|---|
| ☐ | ☒ | The exact sample size (*n*) for each experimental group/condition, given as a discrete number and unit of measurement |
| ☐ | ☒ | A statement on whether measurements were taken from distinct samples or whether the same sample was measured repeatedly |
| ☐ | ☒ | The statistical test(s) used AND whether they are one- or two-sided<br>*Only common tests should be described solely by name; describe more complex techniques in the Methods section.* |
| ☒ | ☐ | A description of all covariates tested |
| ☐ | ☒ | A description of any assumptions or corrections, such as tests of normality and adjustment for multiple comparisons |
| ☐ | ☒ | A full description of the statistical parameters including central tendency (e.g. means) or other basic estimates (e.g. regression coefficient) AND variation (e.g. standard deviation) or associated estimates of uncertainty (e.g. confidence intervals) |
| ☐ | ☒ | For null hypothesis testing, the test statistic (e.g. $F$, $t$, $r$) with confidence intervals, effect sizes, degrees of freedom and $P$ value noted<br>*Give P values as exact values whenever suitable.* |
| ☒ | ☐ | For Bayesian analysis, information on the choice of priors and Markov chain Monte Carlo settings |
| ☒ | ☐ | For hierarchical and complex designs, identification of the appropriate level for tests and full reporting of outcomes |
| ☒ | ☐ | Estimates of effect sizes (e.g. Cohen's *d*, Pearson's *r*), indicating how they were calculated |

*Our web collection on statistics for biologists contains articles on many of the points above.*

## Software and code

Policy information about availability of computer code

| Data collection | FACS Diva 9.0 |
|---|---|
| Data analysis | GraphPad Prism v9.02, Flowjo v10.4.1, R 3.5.0, Seurat (v3.1.5), DESeq2 (v1.24.0) TopHat (v2.09) for human data, TopHat (v1.4.1) for murine data, FastQC (v0.11.2), Bowtie (v.1.1.2), Samtools (v0.1.19.0), HTSeq framework (v0.7.1), Trimmomatic (v.0.36), ggplot2(v3.3.2), data.table (v1.13.2), RColorBrewer (v1.1.2), MAST (v1.10.0), SCDE (v1.99.1), python (v3.8.5), graphviz (v0.8.2), fgsea (v1.10.1), Cell Ranger (v3.1.0), bcl2fastq (v2.20.0.422), uwot (v0.1.8), cowplot (v1.0.0) |

For manuscripts utilizing custom algorithms or software that are central to the research but not yet described in published literature, software must be made available to editors/reviewers. We strongly encourage code deposition in a community repository (e.g. GitHub). See the Nature Research guidelines for submitting code & software for further information.

## Data

Policy information about availability of data

All manuscripts must include a data availability statement. This statement should provide the following information, where applicable:

- Accession codes, unique identifiers, or web links for publicly available datasets
- A list of figures that have associated raw data
- A description of any restrictions on data availability

Raw data are publicly available and GEO links are provided.

# Field-specific reporting

Please select the one below that is the best fit for your research. If you are not sure, read the appropriate sections before making your selection.

☒ Life sciences ☐ Behavioural & social sciences ☐ Ecological, evolutionary & environmental sciences

For a reference copy of the document with all sections, see nature.com/documents/nr-reporting-summary-flat.pdf

# Life sciences study design

All studies must disclose on these points even when the disclosure is negative.

| | |
|---|---|
| Sample size | Murine studies: Sample sizes were chosen based on published studies to ensure sufficient numbers of mice in each group enabling reliable statistical testing and accounting for variability. Sample sizes are indicated in Figure legends. Clinical study: The sample size was calculated as follows: in a pilot cohort, the CD8 count in the biopsy taken at diagnosis, and in the resected tissue sample was quantified. The mean value at diagnosis was 25 cells/high power filed (hpf), and this remained almost the same in the resected sample (26 cells/hpf). With an observed standard deviation of five cells we posited we would observe a doubling to 50 cells/hpf following treatment with AMG 319, hence a difference between the two treatment groups of 25. To detect a standardised difference of 0.5 with 80% power and one sided test of statistical significance of 20%, we required 36 patients to be randomised to AMG319 and 18 to placebo (54 in total). |
| Data exclusions | RNA-seq samples that didn't pass quality control weren't included in the analyses. |
| Replication | Murine data were reliably reproduced in independent experiments at least twice. |
| Randomization | Murine studies: Animals of same sex and age were randomly assigned to experimental groups. Clinical study: Randomisation was at the level of the individual patient, using block randomisation with randomly varying block sizes. During the course of the clinical trial the randomisation list was held by the unblinded Trial Statistician and within the IWRS |
| Blinding | Murine studies: Blinding was not performed. The employed methods involve unbiased quantification (flow cytometry, gene expression). The experimental obervation (change in tumor volume) following therapeutic intervention was substantial and consistent within the respective groups. Hence, the data presented did not require blinding. Clinical study: During the course of the clinical trial the randomisation list was held by the unblinded Trial Statistician and within the IWRS. Patients and care providers were blinded to the treatment allocation, and all immunological evaluations were completed by a pathologist and researchers who were blinded to the patient allocation to treatment arms. |

# Reporting for specific materials, systems and methods

We require information from authors about some types of materials, experimental systems and methods used in many studies. Here, indicate whether each material, system or method listed is relevant to your study. If you are not sure if a list item applies to your research, read the appropriate section before selecting a response.

## Materials & experimental systems

| n/a | Involved in the study |
|---|---|
| ☐ | ☒ Antibodies |
| ☐ | ☒ Eukaryotic cell lines |
| ☒ | ☐ Palaeontology |
| ☐ | ☒ Animals and other organisms |
| ☐ | ☒ Human research participants |
| ☐ | ☒ Clinical data |

## Methods

| n/a | Involved in the study |
|---|---|
| ☒ | ☐ ChIP-seq |
| ☐ | ☒ Flow cytometry |
| ☒ | ☐ MRI-based neuroimaging |

# Antibodies

| | |
|---|---|
| Antibodies used | The following antibodies for floytometry were used in this study.<br><br>Fixable Viability Dye, ThermoFisher Scientific, eFluor780<br><br>anti-human<br><br>Antibody, Supplier, Clone, Color<br><br>PD-1, BD Biosciences, EH12.1, BV421, 1:30, cat#562516<br>CD45, BD Biosciences, HI30, APC-R700, 1:30, cat#566041<br>CD25, BD Biosciences, M-A251, PE, 1:20, cat#555432<br>CD127, eBioscience, eBioRDR5, APC, 1:50, cat#17-1278-42<br>CD4, Biolegend, OKT4, BV510, 1:30, cat#317444<br>CD137, Biolegend, 4B4-1, BV605, 1:30, cat#309822 |

GITR, Biolegend, 108-17, BV711, 1:30, cat#371212
ICOS, Biolegend, C398.4A, BV786, 1:50, cat#313534
CD8A, Biolegend, SK1, PerCP-Cy5.5, 1:30, cat#344710
CD3, Biolegend, SK7, PE-Dazzle594, 1:30, cat#344844
CD14, Biolegend, HCD14, APC-Cy7, 1:50, cat#325620
CD20, Biolegend, 2H7, APC-Cy7, 1:50, cat#302314

anti mouse
ST2, BD Biosciences, U29-93, BV421, 1:100, cat#145309
CD8, BD Biosciences, 53-6.7, BB700, 1:100, cat#566409
Ki67, BD Biosciences, B56, BV786, 1:40, cat#563756
CD45, BD Biosciences, 30-F11, APC-R700, 1:100, cat#565478
TOX, Miltenyi, REA473, APC, 1:40, cat#130-118-335
FOXP3, eBioscience, FJK-16s, APC, 1:100, cat#17-5773-82
CD3, Biolegend, 145-2C11, BV510, 1:100, cat#100353
CD4, Biolegend, RM4-5, PE-Dazzle594, 1:100, cat#100566
PD-1, Biolegend, 29F1.A12, BV605, 1:100, cat#135220
CD19, Biolegend, 6D5, BV650, 1:100, cat#115541
GzmB, Biolegend, QA16A02, PE, 1:40, cat#372208

Validation | All anti-human and anti-mouse antibodies have been validated by the respective manufacturer for flow cytometry.

# Eukaryotic cell lines

Policy information about cell lines

Cell line source(s) | B16F10-OVA cells were a gift from the laboratory of Prof. Linden (LJI)

Authentication | B16F10-OVA cells form distinct melanoma tumors and are thus true melanoma cells. Cell lines were not further authenticated

Mycoplasma contamination | Cell lines tested negative for mycoplasma infection and were subsequently treated with Plasmocin to prevent contamination

Commonly misidentified lines
(See ICLAC register) | No commonly misidentified cell lines were used

# Animals and other organisms

Policy information about studies involving animals; ARRIVE guidelines recommended for reporting animal research

Laboratory animals | C57BL/6J (JAX stock #000664), OT-I (JAX stock #003831), Rag1-/- (JAX stock #002216) and CD8-/- (JAX stock #002665) mice were obtained from Jackson labs. Foxp3RFP (JAX stock #008374) were a kind gift from K. Ley (LJI). Age (6-12 weeks) and sex-matched mice were used for all experiments. The housing temperature is controlled, ranging from 69-75F, humidity is monitored but not controlled and ranges from 30-70%. The light/dark cycles are from 6am-6pm, respectively. All animal work was approved by the relevant La Jolla institute for Immunology Institutional Animal Care and Use Committee.

Wild animals | No wild animals were used

Field-collected samples | No Field-collected samples were used

Ethics oversight | All animal work was approved by the relevant LJI Animal Ethics Committee

Note that full information on the approval of the study protocol must also be provided in the manuscript.

# Human research participants

Policy information about studies involving human research participants

Population characteristics | Newly diagnosed, untreated patients with histologically confirmed HNSCC were prospectively recruited. All patients had tissue collected as a dedicated research biopsy after consent and prior to randomization, with an additional sample collected during surgical resection. All patient characteristics can be found in the Source Data _Patient characteristics

Recruitment | Patients were recruited at four institutions in the UK (University Hospital Southampton NHS Foundation Trust, Poole Hospitals NHS Foundation Trust, Liverpool University Hospitals NHS Foundation Trust and Queen Elizabeth University Hospital Glasgow) and written informed consent was obtained from all subjects. Patients were randomly assigned to either a placebo group or a drug treatment-group. Detailed information about the trial design, randomization procedure, protocol amendments, recruitment data, patient characteristics and adverse events are deposited at https://www.clinicaltrialsregister.eu/ctr-search/trial/2014-004388-20/results#moreInformationSection. Patients were recruited after initial diagnosis and before definitive surgical treatment; drug treatment or placebo was given for up to 24 or 28 days respectively, prior to resection of tumor

| Ethics oversight | The study was sponsored by Cancer Research UK Center for Drug Development (CRUKD/15/004) and approved by the Southampton and South West Hampshire Research Ethics Board; the trial EudraCT number is 2014-004388-20 |

Note that full information on the approval of the study protocol must also be provided in the manuscript.

# Clinical data

Policy information about clinical studies

All manuscripts should comply with the ICMJE guidelines for publication of clinical research and a completed CONSORT checklist must be included with all submissions.

| Clinical trial registration | https://www.clinicaltrialsregister.eu/ctr-search/trial/2014-004388-20/results |
| Study protocol | Summary data, including toxicity listings and a list of protocol amendments have been reported on the EU clinical trials register |
| Data collection | Patients were recruited from October 2015 to May 2018 at in the UK (University Hospital Southampton NHS Foundation Trust, Poole Hospitals NHS Foundation Trust, Liverpool University Hospitals NHS Foundation Trust and Queen Elizabeth University Hospital Glasgow, two additional centers did not recruit patients); written informed consent was obtained from all subjects. Patients were eligible if they were ≥18 years, with histologically proven HNSCC for whom surgery was the primary treatment option, with laboratory results within specified ranges. Patients had to be clinically eligible for tumor resection; patients who had undergone prior radio/immuno/chemotherapy or other anti-cancer therapy for their current HNSCC, were excluded. Clinical data were obtained for age, gender, tumour size (T stage), and nodal status (N stage) (summarised in Source Data_Patient characteristics) |
| Outcomes | Primary endpoints were safety and assessment of CD8+ immune infiltrates, secondary endpoints tumor responses and AMG319 pharmacokinetic evaluation (https://www.clinicaltrialsregister.eu/ctr-search/trial/2014-004388-20/results#endPointsSection). |

# Flow Cytometry

## Plots

Confirm that:

☒ The axis labels state the marker and fluorochrome used (e.g. CD4-FITC).

☒ The axis scales are clearly visible. Include numbers along axes only for bottom left plot of group (a 'group' is an analysis of identical markers).

☒ All plots are contour plots with outliers or pseudocolor plots.

☒ A numerical value for number of cells or percentage (with statistics) is provided.

## Methodology

| Sample preparation | Murine samples - Lymphocytes were isolated from spleen by mechanical dispersion through a 70-μm cell strainer (Miltenyi) to generate single-cell suspensions. RBC lysis (Biolegend) was performed to remove red blood cells. Tumor samples were harvested and lymphocytes were isolated by dispersing the tumor tissue in 2ml of PBS, followed by incubation of samples at 37°C for 15min with 800U/ml DNase I (Sigma) and 0.15WU/ml Liberase DL (Roche). Solutions were then diluted with MACS buffer and grinded through a 70-μm cell strainer to generate a single cell solution

Human samples - Human TILs were isolated from cryopreserved tumor tissue using a combination of enzymatic and mechanical dissociation. TILs were isolated by dispersing the tumor tissue in 1ml of PBS, followed by incubation of samples at 37°C for 15min with 800U/ml DNase I (Sigma) and 0.15WU/ml Liberase DL (Roche). Solutions were then diluted with MACS buffer and grinded through a 70-μm cell strainer to generate a single cell solution |
| Instrument | BD LSRFortessa, BD FACSAria-4 Fusion |
| Software | BD FACSDiva 9 <br> FlowJo v10.4.1 |
| Cell population abundance | Sorting efficiency was observed during sorting |
| Gating strategy | Gating strategy provided in  ED Fig. 2 and ED Fig. 3. |

☒ Tick this box to confirm that a figure exemplifying the gating strategy is provided in the Supplementary Information.

