## [Peer Review File · Nature]

Manuscript Title: Intermittent PI3K δ inhibition sustains anti-tumor immunity and curbs irAEs

Reviewer Comments & Author Rebuttals

Reviewer Reports on the Initial Version:

Referees' comments:

Referee #1 (Remarks to the Author):

This paper is an interesting evaluation of the effect of PI3K delta inhibition in solid tumors. This neoadjuvant trial in HNSCC led to significant irAEs that resulted in the stopping of the trial but the authors have extensive correlative studies looking at immunomodulation. As a general comment, many of the figures could be better labeled or described, and some of the claims may be stronger than is fully justified by the data.

1) While the effects here are different than seen in the heavily pretreated patients on the phase 1 in heme malignancies, they are not dissimilar from the rapid onset severity seen in the untreated CLL patients in Lampson et al. also even in the phase 1, there were irAEs, they were just later and less severe than in more immunocompetent patients. So I would tone down the comments that imply that these results are a phase change, as they are more in a continuum.

2) This paper is really focused on the clinical trial so it might make more sense reorganized to present the initial trial findings and correlatives, then talk about the mice, and then return to the scRNA seq

3) The presentation of the trial results needs to be improved in many places. It should be clear that the planned dose was 400 mg (i.e the way it is written now sounds a bit like a phase 1 dose escalation). The timing of discontinuations should be presented early on p 7 in the initial description of the trial. Extended data table 3 is referred to extensively for detailed patient level data but it is very hard to understand; each worksheet should have each patient labeled as to whether they discontinued for what irAE and on what day of therapy (i.e. right next to the PK data, the AKT data, etc).

4) Figure 3b is not adequately explained, as at first glance, it is not clear why anti-IgD is used for determining pAKT MFI. Also the 3 patients with objective response should have a separate label in all figures.

5) Figure 4c does not have as much information as desired. A swimline plot with each patient indicated, showing the timing and nature of their toxicity and the timing of their discontinuation or completion of study drug, should be shown for study drug and placebo patients.

6) Changes in circulating Tregs in the patients should be assessed in relation to patients who did or did not have toxicity, as well as in relation to severity of toxicity. Also the data on the placebo group should also be shown.

7) P. 8, the authors state that the increase in circulating Tregs reflects displacement from tissues. But how do they know that? If they have tissue biopsies they should show them. If the data are based on

analogy to the mice, then they should say that, and this could be a place where the mouse data would fit more seamlessly in the paper.

8) P. 9 the RNA seq is done on pre and post treatment tumor samples. When exactly were these post treatment tumor samples done in relation to drug dosing vs discontinuation? This should be specified in all cases.

9) Figure 6c is inscrutable and requires more explanation

10) The bulk rna seq data showing increased GZMA, they don't know that this is driven by CD8 T cells, could also be expressed in Tregs.

11) P 10 refers to intermittent dosing of AMG319 – was this intermittent? If so that should be stated upfront.

12) The claims regarding enhanced CD4 and CD8 activation in the TME, and decrease in tumor-infiltrating Tregs would be supported by some IHC data. How do they reconcile this with the low TILs observed? And the decrease in tumor-infiltrating Tregs in the human tissue is inferred, but that is not clearly stated.

13) The final sc RNA sequencing data from 6 pts: what about looking at FOXP3 positive cells in this to assess the impact on Tregs here?

Referee #2 (Remarks to the Author):

In their paper, Eschweiler et al aim at exploring the immune effects of PI3K δ i.

In general, the impact of this class of drug on regulatory T cells has already been reported extensively. Most of the data reported in this paper add therefore minimal information to previous knowledge, and do not bring new scientific concept.

The figure 1 does not provide evidence that the immune modulation induced by PI-3065 leads to tumor rejection. To address this point, the authors should show that the drug does not have antitumor effect in an immunocompromised mice model.

Single cell experiments should be used to generate hypotheses, that should further be proven using standard methods and predefined hypotheses. In the present paper, while the single cell experiments are nicely done and generate interesting hypotheses, they are usually not followed by confirmatory experiments. For this reason, the paper lacks focus and addresses too many questions. The authors should pick a data generated by single cell analyses and go deep in the validation. As example, authors could investigate deeper the impact of PI3K δ i on the population 2 and 8 (Figure 2), since these are quite new data.

The clinical part is extremely difficult to interpret because of the small sample size and the way data are reported. Overall, the authors do not report strong evidence that PI3K δ i decreases regulatory T cells since statistical tests does not reach low p value. The way the clinical trial and biomarker analyses are reported should be improved. As example, they should comply with CONSORT and REMARK methods of reporting.

The finding that PI3K δ i induce colitis has already been reported. The magnitude of toxicity then depends on the dose used in the trial, and the disease itself. Since patients presenting lymphoproliferative disorders are immunocompromised, it makes sense that patients with head and neck cancer present more toxicity. I think this observation suggests that the dose selected in phase I trials is not the right one.

It seems the single cell analyses pre- / post- treatment are done only in patients included in the AMG arm. Patients from the placebo arm should be profiled as control because biopsy itself induces lymphocytic infiltration. How the six patients were selected ?

Minor comments :

There are many sentences with approximation (« suggest » , « presumably by » , « trend (p=0.06 » ...)

Referee #3 (Remarks to the Author):

The manuscript by Eschweiler et al first describes the effects of a PI3Kdelta inhibitor in mouse models of cancer and then analysis of a neoadjuvant clinical trial of a PI3Kdelta inhibitor in HNSCC patients. PI3Kdelta has been shown (best by the group of Bart Vanhaesbroeck, a co-author on this manuscript) to be an important signaling molecule for Treg expansion and suppressive function. Thus, a drug inhibitor specific for this PI3K isoform is an interesting potential small molecule therapeutic given the evidence in mice that Treg play an important role in suppressing anti-tumor immunity. There are currently no Treg-specific drugs in testing.

There are a number of issues that weaken the paper.

Murine data aimed at studying effects of PI3Kdelta inhibitor on anti-tumor immune cells:

- Given that the PI3Kdelta KO mice have been previously analyzed to show that Treg are decreased and defective and fail to inhibit anti-tumor immunity, resulting in tumor growth inhibition, and given that the leader of that study is a co-author of the current manuscript, it seems that the authors missed the opportunity to compare the cellular composition and transcriptomics of the inhibitor treated mice with the KO mice to determine how well the inhibitor phenocopies the KO. There is no information on off-target TK inhibition with this drug, which, together with the comparison to KO mice, would give us a better sense of how the drug is working at the in vivo immunologic level.
 - In the original KO paper, B16 was used whereas in this manuscript, B16ova, which is far more immunogenic, was used. As per my comment above, using B16 and starting the drug at time of implantation would be the ideal comparison with previously published KO data. Using B16-ova, they could have taken the opportunity to evaluate effects on ova-specific T cells by doing adoptive transfer of OT-I and OT-II cells.
 - While there are interesting changes in the colonic Tregs on sc analysis, the Treg in tumor do not look like they are much affected at all by the drug. Wouldn't this be the most important population considering the clinical analysis is largely about effects of a PI3Kdelta inhibitor on tumor Treg?
 - The authors suggest that cluster 8 in the colonic Treg is the equivalent of the "ST2 Treg" but do not show ST2 levels.
 - Is there colitis in the inhibitor treated mice?
 - Seems the most prominent shift with inhibitor is from cluster 2 to cluster 5 but the RNA expression profile differences are not assessed between these two clusters. Instead they show a few violin plots of some genes comparing cluster 0 to cluster 8. Not sure why they show this or what it adds.
- Clinical trial:
- They report the trial as inconclusive based on histology but do not give much detail about the

histologic responses. There is a reasonable literature on pathologic responses to neoadjuvant anti-PD-1 in lung cancer, melanoma and HNSCC, so it would be interesting to know more about the PRs and CR that were seen in the treatment group and if there was anything that stood out in the path responders at least as potential signal. Also, given the number of patients who developed significant rashes, biopsy with IHC for relevant T cell subsets and other molecules would have been informative.

- Fundamentally, they do not find any significant effects on intra-tumoral Treg numbers in PI3Kdelta inhibitor treated patients vs control. While there is a small decrease in Foxp3 mRNA on bulk TCRseq, there is NO statistical difference by Foxp3 IHC, which is a much better indicator of Treg because there are many post-transcriptional mechanisms that regulate Foxp3 protein; thus, protein, not RNA is critical. I was surprised they did not do Foxp3 by FACS, given that one gets a lot of T cells out of a HNSCC resection and even out of pre-treatment biopsies if one can get a few passes with an 18 gauge biopsy needle. They do find an increase in Treg in the peripheral blood but unclear what this means given the absence of significant Treg in the tumor. They conclude that the drug pushes Treg out of the tissue and into the peripheral blood – the dynamics of leukocytes between tissues and blood is complicated and one cannot draw conclusions from simply finding changes in number within a given compartment without employing some sort of tracking approach, which is often done in mouse models but is very difficult in humans.
- The analysis of transcriptional programs of Treg, in particular genes known to regulate Treg function, and genes that the authors indicate are regulated by PI3K, are not evaluated. The Treg cells are not identified in the sc analysis in Fig 6a and the violin plots in Fig 6b (they refer to Fig 5b in the text but I think this is a mistake) only look at genes associated with CTL function. Visually, there does not appear to be much of a difference with inhibitor vs control and I did not see any statistics – they also refer to supplementary table 5 but that table is TCRseq. Figure 6c is incomprehensible and 6d doesn't tell us much. Without knowing the specificity of the TCRs, hard to know what the modest clonal expansions ultimately mean.

Author Rebuttals to Initial Comments:

Point-by-point plan for addressing reviewer's comments (authors responses and changes to manuscript in blue font).

Referee #1 (Remarks to the Author):

This paper is an interesting evaluation of the effect of PI3K delta inhibition in solid tumors. This neoadjuvant trial in HNSCC led to significant irAEs that resulted in the stopping of the trial but the authors have extensive correlative studies looking at immunomodulation. As a general comment, many of the figures could be better labelled or described, and some of the claims may be stronger than is fully justified by the data.

We thank the reviewer for recognizing the importance of our study and for the very positive assessment in elucidating the immunomodulatory effects of PI3K δ inhibition in solid tumors. As suggested, we have added additional labels to the figures and provide data to support our claims.

1) While the effects here are different than seen in the heavily pretreated patients on the phase 1 in hematologic malignancies, they are not dissimilar from the rapid onset severity seen in the untreated CLL patients in Lampson et al. also even in the phase 1, there were irAEs, they were just later and less severe than in more immunocompetent patients. So I would tone down the comments that imply that these results are a phase change, as they are more in a continuum.

We thank the reviewer for this careful comment and agree that the development of irAEs was not entirely unexpected. We merely tried to highlight that the timing and at times, severity, at which these irAEs occurred in our patient cohort, were very surprising. These important findings will have to be considered in potential follow-up studies to better balance the immunomodulatory effects of PI3K δ inhibition in immunocompetent patients while minimizing immune related toxicity. As suggested by this reviewer, we now have toned down the language in the pertaining section to more accurately reflect the nuances of PI3K δ i-mediated irAEs with regard to their description in previous studies.

Crucially however, we now provide additional data outlining the immunomodulatory effects of PI3K δ i, delineating why it causes irAEs in non-malignant organs.

“To explore the connection between PI3K δ inhibition and gastrointestinal toxicity in more detail, we utilized a Dextran Sulfate Sodium (DSS)-induced acute colitis model. Crucially, when compared to placebo-treated mice, we found that PI3K δ inhibition led to an accelerated and exacerbated disease phenotype, with a swift reduction in body weight and a higher overall colitis score characterized by significantly higher inflammation, crypt damage and area of infiltration (extent) (Fig. 3a,b), indicative of treatment-mediated alterations in tissue homeostasis driving immunopathology.” – Page 10

“To discern whether specific T cell subsets drive immunopathology upon PI3K δ inhibition, we performed scRNA sequencing of colonic CD8⁺ and CD4⁺ T cells in the different treatment regimens... Strikingly, we found a dosing-dependent enrichment of the T_C17 and T_H17 subsets and pertaining proliferating clusters, making up ~50% of all cells in the continuous dosing regimen, while they were nearly completely absent in the other treatment conditions (Fig. 4a and Extended Data Fig. 6a). Importantly, IL-17 producing cells have been shown to cause colitis^{41–43}. Cells in these IL-17⁺ clusters were moreover heavily clonally expanded and exhibited substantial cellular and clonotypic overlap in both CD8⁺ and CD4⁺ T cells (Fig. 4d,e, Extended Data Fig. 6d,e and Extended Data Table 4), likely contributing to their rapid expansion. Conversely, we found a dosing-dependent decrease of innate-like CD8⁺ T cells, which have been implicated in controlling inflammation and the onset of colitis^{44,45} (Fig. 4a-c).” – Page 11.

2) This paper is really focused on the clinical trial so it might make more sense reorganized to present the initial trial findings and correlatives, then talk about the mice, and then return to the scRNA-seq

We thank this reviewer for this excellent suggestion and have altered both the order and presentation of the data. Given the amount and far-reaching implications of the pivotal murine data we now provide, we have honed in on the aspect of immune related toxicity observed in the clinical trial participants and now present extensive data pointing to a possible mode of action.

3) The presentation of the trial results needs to be improved in many places. It should be clear that the planned dose was 400 mg (i.e the way it is written now sounds a bit like a phase 1 dose escalation). The timing of discontinuations should be presented early on p 7 in the initial description of the trial. Extended data table 3 is referred to extensively for detailed patient level data but it is very hard to understand; each worksheet should have each patient labeled as to whether they discontinued for what irAE and on what day of therapy (i.e. right next to the PK data, the AKT data, etc).

We thank the reviewer for this comment and apologize if the trial results were not clearly depicted in the figures. We have revised the figures and worksheets describing the clinical data to improve the overall presentation of our findings.

4) Figure 3b is not adequately explained, as at first glance, it is not clear why anti-IgD is used for determining pAKT MFI. Also the 3 patients with objective response should have a separate label in all figures.

We apologize if this was not clearly explained. We utilized anti-IgD stimulation and activate B cells in order to measure the target inhibition (pAKT levels) in B cells and further highlight this in the methods section. Appropriate labels have been added to the figures. We now also highlight the 3 patients in whom we observed either partial or complete responses in Fig. 1b.

5) Figure 4c does not have as much information as desired. A swimline plot with each patient indicated, showing the timing and nature of their toxicity and the timing of their discontinuation or completion of study drug, should be shown for study drug and placebo patients.

We thank this reviewer for this excellent suggestion and agree that this would significantly strengthen the visualization of the clinical data. We now provide the swimline plot depicting treatment intervals, as well as time points and grades of irAEs (Fig. 1b).

6) Changes in circulating Tregs in the patients should be assessed in relation to patients who did or did not have toxicity, as well as in relation to severity of toxicity. Also the data on the placebo group should also be shown.

As requested, we have now placed the data on Placebo- and PI3Kdi-treated patients next to one another to improve visibility (Extended Data Fig. 3d,e).

7) P. 8, the authors state that the increase in circulating Tregs reflects displacement from tissues. But how do they know that? If they have tissue biopsies they should show them. If the data are based on analogy to the mice, then they should say that, and this could be a place where the mouse data would fit more seamlessly in the paper.

We thank the reviewer for raising this important point as it pertains to a crucial part of the manuscript. As we did not anticipate the early onset and severity of irAEs in this study, we did not have ethical permission to obtain tissue biopsies and had thus no way to investigate this hypothesis in greater detail. As inferred by this reviewer, we derive to our conclusion based on findings in published studies (Luo et al. 2016 and reviewed in Johansen et al. 2021) and based on the results in our preclinical models, in which we observed significant T_{REG} cell depletion in all assessed organs. We have moreover framed this hypothesis more carefully now. Please also see our response to points #1 and #6.

As suggested by the reviewer, we have now modified the results section

*“Interestingly, PI3K δ -inhibition led to a significant increase in activated circulating T_{REG} cells, while the proportion and activation status of T_{REG} cells in the placebo group remained stable (**Extended Data Fig. 3d,e**). This implies that PI3K δ inhibition either influences proliferation, or that it displaces activated T_{REG} cells from tissues, presumably by altering the expression of tissue homing factors like KLF2 and S1PR1, direct targets of FOXO1 in line with previous studies⁵⁻⁷, likely contributing to toxicity.” – Page 6, 7*

8) P. 9 the RNA seq is done on pre and post treatment tumor samples. When exactly were these post treatment tumor samples done in relation to drug dosing vs discontinuation? This should be specified in all cases.

We apologize for not stating this in the initial version of the manuscript. We have now added which patient samples were used for the scRNA-seq in the figure legend of **Extended Data Fig. 3a** and the dosing regimen for all patients is now clearly depicted in **Fig. 1b**.

9) Figure 6c is inscrutable and requires more explanation

We apologize for not explaining these data in greater detail. This figure depicted the level of clonal expansion in $CD4^+$ and $CD8^+$ T cells respectively and pertained to **Fig. 6a** in the previous version of the manuscript. We realize that the visualization was not ideal and have thus removed this figure from the manuscript. The same data are depicted in the bar charts in **ED Fig.3c** in the new version.

10) The bulk rna seq data showing increased GZMA, they don't know that this is driven by $CD8^+$ T cells, could also be expressed in Tregs.

We thank this reviewer for this careful comment and agree, that the increase in GZMA could theoretically stem from T_{REG} cells. To avoid ambiguity, we now emphasize our analysis of purified $CD8^+$ T cells and single-cell RNA-seq analysis of tumor-infiltrating $CD8^+$ T cells to support the conclusion of their enhanced cytotoxic potential following PI3K δ inhibitor treatment.

11) P 10 refers to intermittent dosing of AMG319 – was this intermittent? If so that should be stated upfront.

We apologize for not depicting treatment duration and intervals in the initial version of the manuscript. The occurrence of irAEs prevented uninterrupted treatment, as is now clearly depicted in **Fig. 1b**.

12) The claims regarding enhanced $CD4^+$ and $CD8^+$ activation in the TME, and decrease in tumor-infiltrating Tregs would be supported by some IHC data. How do they reconcile this with the low TILs observed? And the decrease in tumor-infiltrating Tregs in the human tissue is inferred, but that is not clearly stated.

As mentioned in the manuscript, we focused on Human Papilloma Virus (HPV)-negative HNSCC, as this cancer type is more prevalent, and because patients with this cancer type have poorer outcomes when compared to HPV-positive HNSCC, likely due to overall lower tumor infiltrating lymphocyte (TIL) infiltration. Unfortunately, this low TIL infiltration was the very reason why we could not analyze potential alterations in tumor-infiltrating T_{REG} cell numbers in greater detail or reliability. However, we do find a reduction in *FOXP3* transcript levels in the tumor samples, indicative of lower T_{REG} cell numbers in the tumor. Moreover, the increase in cytotoxicity of $CD4^+$ and $CD8^+$ T cells were independently verified in both bulk RNA-seq (**Extended Data Fig. 2c**) and single cell RNA-seq (**Extended Data Fig. 3b**) and are also in line with our findings in murine tumors (**Extended Data Fig. 4b-e**).

13) The final sc RNA sequencing data from 6 pts: what about looking at FOXP3 positive cells in this to assess the impact on Tregs here?

We thank this reviewer for this comment. We had indeed hoped to analyse FOXP3⁺ T cells in greater detail with this approach and thus relied on plate-based smart-seq2. To accommodate FOXP3⁺ T cells in this approach, we sorted CD3⁺ T cells from tumor samples pre- and post-treatment. Unfortunately, however, after removal of low-quality cells, we ended up with very few CD4⁺FOXP3⁺ T cells (0-27 cells per patient) from the initially sequenced 2,342 CD3⁺ T cells, precluding any in-depth analysis and conclusions on this cell type. Please also see our response to point #12.

Referee #2 (Remarks to the Author):

In their paper, Eschweiler et al aim at exploring the immune effects of PI3K δ i. In general, the impact of this class of drug on regulatory T cells has already been reported extensively. Most of the data reported in this paper add therefore minimal information to previous knowledge, and do not bring new scientific concept.

Based on excellent suggestions by this reviewer, we have now included extensive data that bring new scientific concepts regarding mechanisms of immune-related adverse events (irAEs) and anti-tumor immune responses driven by PI3K δ inhibition.

15) The figure 1 does not provide evidence that the immune modulation induced by PI-3065 leads to tumor rejection. To address this point, the authors should show that the drug does not have antitumor effect in an immunocompromised mice model.

We thank this reviewer for this comment and agree, that the alterations in tumor growth could stem from cancer cell-intrinsic effects of PI3K δ inhibition as has been shown for B cell malignancies. We have therefore performed additional experiments in both RAG1^{-/-} and CD8^{-/-} knockout mice. These new results clearly demonstrate the immunomodulatory effects of PI3K δ inhibition and exclude potential cancer cell-intrinsic effects.

“Given that PI3K inhibitors were initially considered to mainly target cancer cell-intrinsic PI3K activity, we verified that the observed anti-tumor effects were dependent on immune cells, and more specifically on CD8⁺ T cells (Extended Data Fig. 4h).” – Page 7

16) Single cell experiments should be used to generate hypotheses, that should further be proven using standard methods and predefined hypotheses. In the present paper, while the single cell experiments are nicely done and generate interesting hypotheses, they are usually not followed by confirmatory experiments. For this reason, the paper lacks focus and addresses too many questions. The authors should pick a data generated by single cell analyses and go deep in the validation. As example, authors could investigate deeper the impact of PI3Kdi on the population 2 and 8 (Figure 2), since these are quite new data.

We thank the reviewer for this valid point. We have now restructured the manuscript and performed several additional experiments to hone in on the emergence and mode of action of irAEs. These pivotal new data demonstrate that PI3K δ inhibition causes colitis and moreover delineate the precise mechanism driving the accumulation of pathogenic T cell compartments in colonic tissue. We now explore how alterations in colonic T_{REG} cell subtypes contribute to disease phenotype and irAEs.

“While colonic T_{REG} cells in cluster 0 and cluster 8 shared this ST2 signature (Fig. 2e), only cells in cluster 8 showed high transcript expression of the immunosuppressive cytokine IL-10 (Fig. 2f). Moreover, cells in cluster 8 were also enriched for transcripts linked to cellular activation (Cd44, Icos and Klrg1) and superior suppressive capacity (Ctla4, and Gzmb) (Extended Data Fig. 5g). These T_{REG} cell clusters (2 and 8) with highly suppressive properties were depleted in PI3K δ i-treated mice, while the clonally expanded cluster 5 T_{REG} cells were enriched in PI3K δ i-treated mice showed a lack of transcript associated with suppression (Fig. 2f and Extended Data Fig. 5g) and instead higher expression of several interferon-related response genes (Stat1, Stat3, Ifrd1)^{37,38}, suggestive of a pro-inflammatory environment (Fig. 2d). Accordingly, ST2⁺Il10⁺ T_{REG} cells were substantially

reduced in PI3K δ i-treated mice (**Fig. 2g**). Interestingly, RNA velocity analysis, a tool to assess the developmental stage of cells in scRNA-seq data^{39,40}, infers a developmental trajectory over several progenitor states (cluster 2,4 and then 0) culminating in clonally expanded ST2 T_{REG} cells (cluster 8) in placebo-treated mice (**Fig. 2a,h**). These data indicate that PI3K δ inhibition prevents the cellular differentiation into ST2 T_{REG} cells, and instead diverts development to cluster 5 T_{REG} cells that lack transcript expression associated with suppressive capacity, pointing to a possible mechanism for the onset of inflammation and colitis.” – Page 9

Please also see our response to point #1 and #2.

17) The clinical part is extremely difficult to interpret because of the small sample size and the way data are reported. Overall, the authors do not report strong evidence that PI3Kdi decreases regulatory T cells since statistical tests does not reach low p value.

We thank this reviewer for this comment and agree that the small sample size precludes firm statements on the effects of PI3K δ inhibition on T_{REG} cells in our human study participants and addressed this concern in the manuscript.

“While immunohistochemistry analysis (IHC) showed a similar trend towards fewer T_{REG} cells (**Extended Data Fig. 2b**), this did not reach statistical significance in the pairwise evaluation, likely because small pre-treatment biopsies precluded accurate cell counts.” – Page 6

However, we do find a reduction in FOXP3 transcript levels in the tumor samples, indicative of lower T_{REG} cell numbers in the tumor. Moreover, the increase in cytotoxicity of CD4⁺ and CD8⁺ T cells were independently verified in both bulk RNA-seq (**Extended Data Fig. 2c**) and single cell RNA-seq (**Extended Data Fig. 3b**) and are also in line with our findings in murine tumors (**Extended Data Fig. 4b-e**).

Please also see our response to point #12.

18) The way the clinical trial and biomarker analyses are reported should be improved. As example, they should comply with CONSORT and REMARK methods of reporting.

We agree that this would strengthen the ease of evaluation of our findings and have thus complied with CONSORT for reporting our data and have attached a CONSORT checklist.

19) The finding that PI3Kdi induce colitis has already been reported. The magnitude of toxicity then depends on the dose used in the trial, and the disease itself. Since patients presenting lymphoproliferative disorders are immunocompromised, it makes sense that patients with head and neck cancer present more toxicity. I think this observation suggests that the dose selected in phase I trials is not the right one.

We thank this reviewer for this valid observation and agree that dosing has to be adjusted for future trials. Importantly, we now provide data supporting this hypothesis and identify intermittent dosing as a promising approach combining sustained anti-tumor immunity with reduced toxicity.

“We tested this hypothesis by utilizing distinct treatment regimens, where mice would either be kept on PI3K δ i for the duration of the experiment (continuous dosing), be kept on PI3K δ i for 4 days followed by 3 days off drug (intermittent dosing) or be kept on PI3K δ i for 2 days followed by 5 days off drug (infrequent dosing) for a total of 2 treatment cycles (**Fig. 3c**). Strikingly, we found that all treatment conditions led to a decrease in tumor growth, albeit not significantly for the infrequent dosing condition, suggesting that transient interruptions of the immunosuppressive TME drive anti-tumor immunity. Most importantly, only continuous dosing led to increased CD8⁺ T cell infiltration and decreased T_{REG} cell levels in colonic tissue (**Fig. 3d**), indicating that intermittent dosing regimens might decrease irAEs in human also.” – Page 10

“Strikingly, we found a dosing-dependent enrichment of the T_C17 and T_H17 subsets and pertaining proliferating clusters, making up ~50% of all cells in the continuous dosing regimen, while they were nearly completely absent in the other treatment conditions (**Fig. 4a** and **Extended Data Fig. 6a**). Importantly, IL-17 producing cells have

been shown to cause colitis⁴¹⁻⁴³. Cells in these IL-17⁺ clusters were moreover heavily clonally expanded and exhibited substantial cellular and clonotypic overlap in both CD8⁺ and CD4⁺ T cells (Fig. 4d,e, Extended Data Fig. 6d,e and Extended Data Table 4), likely contributing to their rapid expansion. Conversely, we found a dosing-dependent decrease of innate-like CD8⁺ T cells, which have been implicated in controlling inflammation and the onset of colitis^{44,45} (Fig. 4a-c).” – Page 11

20) It seems the single cell analyses pre- / post- treatment are done only in patients included in the AMG arm. Patients from the placebo arm should be profiled as control because biopsy itself induces lymphocytic infiltration. How the six patients were selected?

No prior selection was done. Processing was done in all patients with pre and post-AMG treatment tumor samples and only patients with tumor samples from which sufficient number of viable T cells could be FACS-sorted were included for analysis. We did not have sufficient tumor samples from placebo-treated patients to perform single-cell analysis. Furthermore, we did not expect substantial alterations in the transcriptomic signatures in placebo-treated patients, as evidenced by the lack of differentially expressed genes in this patient cohort (3 DEGs in our bulk RNA-seq of pre and post placebo-treated patients, Extended Data Fig. 2a).

Minor comments:

21) There are many sentences with approximation (« suggest » , « presumably by » , « trend (p=0.06 » ...)

These sentences and pertaining data have now been modified or removed from the manuscript.

Referee #3 (Remarks to the Author):

The manuscript by Eschweiler et al first describes the effects of a PI3Kdelta inhibitor in mouse models of cancer and then analysis of a neoadjuvant clinical trial of a PI3Kdelta inhibitor in HNSCC patients. PI3Kdelta has been shown (best by the group of Bart Vanhaesbroeck, a co-author on this manuscript) to be an important signaling molecule for Treg expansion and suppressive function. Thus, a drug inhibitor specific for this PI3K isoform is an interesting potential small molecule therapeutic given the evidence in mice that Treg play an important role in suppressing anti-tumor immunity. There are currently no Treg-specific drugs in testing.

We thank the reviewer for this very positive assessment of the depth and importance of our work.

Major concerns

22) Given that the PI3Kdelta KO mice have been previously analyzed to show that Treg are decreased and defective and fail to inhibit anti-tumor immunity, resulting in tumor growth inhibition, and given that the leader of that study is a co-author of the current manuscript, it seems that the authors missed the opportunity to compare the cellular composition and transcriptomics of the inhibitor treated mice with the KO mice to determine how well the inhibitor phenocopies the KO.

We thank the reviewer for this comment but believe that, while potentially interesting, comparing the single-cell transcriptomic profiles of PI3K δ knockout mice with PI3K δ inhibitor-treated mice is beyond the scope of our manuscript. As the reviewer concedes, the effects of PI3K δ on anti-tumor immunity have already been reported in previous studies. Instead, as suggested by this reviewer (points #24, 25 and 27), we extensively focused on the immunomodulatory effects of PI3K δ inhibition and assessed why and how PI3K δ inhibition causes irAEs, investigated whether altered dosing regimens could reduce toxicity without affecting efficacy. We have now added these new and extensive data to the manuscript.

23) There is no information on off-target TK inhibition with this drug, which, together with the comparison to KO mice, would give us a better sense of how the drug is working at the in vivo immunologic level.

We thank the reviewer for this comment. However, it has been reliably shown that PI-3065 exhibits a >100-fold selectivity over the other isoforms. While this does not rule out potential effects on these kinases, it is likely that the observed effects are indeed facilitated by regulating PI3K δ .

To clarify the alterations in tumor growth could stem from cancer cell-intrinsic effects of PI3K δ inhibition as has been shown for B cell malignancies. We have therefore performed additional experiments in both RAG1 $^{-/-}$ and CD8 $^{-/-}$ knockout mice. These new results clearly demonstrate the immunomodulatory effects of PI3K δ inhibition and exclude potential cancer cell-intrinsic effects.

“Given that PI3K inhibitors were initially considered to mainly target cancer cell-intrinsic PI3K activity, we verified that the observed anti-tumor effects were dependent on immune cells, and more specifically on CD8 $^{+}$ T cells (Extended Data Fig. 4h).” – Page 7

24) In the original KO paper, B16 was used whereas in this manuscript, B16ova, which is far more immunogenic, was used. As per my comment above, using B16 and starting the drug at time of implantation would be the ideal comparison with previously published KO data. Using B16-ova, they could have taken the opportunity to evaluate effects on ova-specific T cells by doing adoptive transfer of OT-I and OT-II cells.

We thank the reviewer for this suggestion. We utilized CD8 $^{-/-}$ mice, adoptively transferred OT-I T cells and treated these tumor-bearing mice with intermittent PI3K δ i (please see Fig. 3c). We have included these data for the reviewer’s discretion (Fig. R1).

Fig. R1: CD8 $^{-/-}$ mice were inoculated with 1.2×10^5 B16F10-OVA cells and treated as indicated. On day 5 after tumor inoculation, 1×10^6 OT-I T cells were adoptively transferred *i.v.* Depicted are tumor volume and the frequency of Ki-67 $^{+}$ and Gzmb $^{+}$ OT-I T cells.

25) While there are interesting changes in the colonic Tregs on sc analysis, the Treg in tumor do not look like they are much affected at all by the drug. Wouldn't this be the most important population considering the clinical analysis is largely about effects of a PI3Kdelta inhibitor on tumor Treg?

We thank the reviewer for this valid comment. While we agree that assessing changes in tumor T_{REG} cell populations or subtypes would be interesting, the alterations seem to be mostly quantitative (Extended Data Fig. 4i). As suggested by the reviewer (please see below, point #26 and 27), we now focus on alterations in colonic T cell compartments (T_{REG}, CD4 $^{+}$ and CD8 $^{+}$ T cells) that drive irAEs. We have added extensive data delineating how and why PI3K δ inhibition leads to colitis and have identified pathogenic T cell populations driving this effect.

“Since gastrointestinal toxicity is one of the major irAEs in patients receiving PI3K δ i^{4,12,13} (Fig. 1b), we hypothesized that T_{REG} cells present in colonic tissue may be especially sensitive to PI3K δ i. To test this hypothesis in an unbiased manner, we performed single-cell RNA-sequencing of T_{REG} cells isolated from tumor, spleen (lymphoid organ) and colonic tissue of PI3K δ i- and placebo-treated B16F10-OVA tumor-bearing FoxP3-RFP reporter mice.” – Page 8

“Interestingly, RNA velocity analysis, a tool to assess the developmental stage of cells in scRNA-seq data^{39,40}, infers a developmental trajectory over several progenitor states (cluster 2,4 and then 0) culminating in clonally expanded ST2 T_{REG} cells (cluster 8) in placebo-treated mice (Fig. 2a,h). These data indicate that PI3K δ inhibition prevents the cellular differentiation into ST2 T_{REG} cells, and instead diverts development to cluster 5 T_{REG} cells that lack transcript expression associated with suppressive capacity, pointing to a possible mechanism for the onset of inflammation and colitis.” – Page 9

“Strikingly, we found a dosing-dependent enrichment of the T_C17 and T_H17 subsets and pertaining proliferating clusters, making up ~50% of all cells in the continuous dosing regimen, while they were nearly completely absent in the other treatment conditions (Fig. 4a and Extended Data Fig. 6a). Importantly, IL-17 producing cells have been shown to cause colitis^{41–43}. Cells in these IL-17⁺ clusters were moreover heavily clonally expanded and exhibited substantial cellular and clonotypic overlap in both CD8⁺ and CD4⁺ T cells (Fig. 4d,e, Extended Data Fig. 6d,e and Extended Data Table 4), likely contributing to their rapid expansion.” – Page 11

26) The authors suggest that cluster 8 in the colonic Treg is the equivalent of the “ST2 Treg” but do not show ST2 levels.

We thank the reviewer for raising this valid point. We have now added these data to the manuscript and also show that PI3K δ inhibition also significantly alters the ratio of ST2 T_{REG} cells to CD8⁺ T cells, likely contributing the emergence of irAEs.

“We verified ST2 expression on T_{REG} cells at the protein level and found that PI3K δ inhibition led to a substantially increased ratio of CD8⁺ T cells to ST2 T_{REG} cells (Extended Data Fig. 5h).” – Page 9

27) Is there colitis in the inhibitor treated mice?

We thank the reviewer for this important question. We have collaborated with Mitchell Kronenberg’s (now a senior co-author), who specialise in colitis models, and performed several critical experiments to answer this very question. We now not only demonstrate that PI3K δ inhibition accelerates and exacerbates colitis in mice, but also unravelled the mechanism causing this.

“To explore the connection between PI3K δ inhibition and gastrointestinal toxicity in more detail, we utilized a Dextran Sulfate Sodium (DSS)-induced acute colitis model. Crucially, when compared to placebo-treated mice, we found that PI3K δ inhibition led to an accelerated and exacerbated disease phenotype, with a swift reduction in body weight and a higher overall colitis score characterized by significantly higher inflammation, crypt damage and area of infiltration (extent) (Fig. 3a,b), indicative of treatment-mediated alterations in tissue homeostasis driving immunopathology.” – Page 10

Please also see our response to point #1 and #25.

28) Seems the most prominent shift with inhibitor is from cluster 2 to cluster 5 but the RNA expression profile differences are not assessed between these two clusters. Instead they show a few violin plots of some genes comparing cluster 0 to cluster 8. Not sure why they show this or what it adds

We thank the reviewer for this comment. We have now added several lines of evidence suggesting that PI3K δ inhibition diverts the development of several progenitor populations into ST2 T_{REG} cells. These pivotal new data imply that cluster 0 T_{REG} cells are direct progenitors of ST2 T_{REG} cells (cluster 8) not yet endowed with substantial suppressive capacity. As such, comparing cluster 0 and cluster 8 T_{REG} cells is of critical importance as it illustrates this very point.

We agree it is also important to highlight the features of clusters 2 and 5, and have included details.

“While colonic T_{REG} cells in cluster 0 and cluster 8 shared this ST2 signature (Fig. 2e), only cells in cluster 8 showed high transcript expression of the immunosuppressive cytokine IL-10 (Fig. 2f). Moreover, cells in cluster 8 were also enriched for transcripts linked to cellular activation (Cd44, Icos and Klrg1) and superior suppressive capacity (Ctla4, and Gzmb) (Extended Data Fig. 5g). These T_{REG} cell clusters (2 and 8) with highly suppressive properties were depleted in PI3K δ -treated mice, while the clonally expanded cluster 5 T_{REG} cells were enriched in PI3K δ -treated mice showed a lack of transcript associated with suppression (Fig. 2f and Extended Data Fig. 5g) and instead higher expression of several interferon-related response genes (Stat1, Stat3, Ifrd1)^{37,38}, suggestive of a pro-inflammatory environment (Fig. 2d). Accordingly, ST2⁺Il10⁺ T_{REG} cells were substantially reduced in PI3K δ -treated mice (Fig. 2g). Interestingly, RNA velocity analysis, a tool to assess the developmental stage of cells in scRNA-seq data^{39,40}, infers a developmental trajectory over several progenitor states (cluster

2,4 and then 0) culminating in clonally expanded ST2 T_{REG} cells (cluster 8) in placebo-treated mice (Fig. 2a,h). These data indicate that PI3K δ inhibition prevents the cellular differentiation into ST2 T_{REG} cells, and instead diverts development to cluster 5 T_{REG} cells that lack transcript expression associated with suppressive capacity, pointing to a possible mechanism for the onset of inflammation and colitis.” – Page 9

29) They report the trial as inconclusive based on histology but do not give much detail about the histologic responses. There is a reasonable literature on pathologic responses to neoadjuvant anti-PD-1 in lung cancer, melanoma and HNSCC, so it would be interesting to know more about the PRs and CR that were seen in the treatment group and if there was anything that stood out in the path responders at least as potential signal. Also, given the number of patients who developed significant rashes, biopsy with IHC for relevant T cell subsets and other molecules would have been informative.

We thank the reviewer for raising this important point. While we agree that these data would be informative and important for future studies, the small sample size and low TIL infiltration preclude drawing reliable conclusions on potential drivers or inhibitors of treatment efficacy. Accordingly, we raise potential conclusions only in the discussion of the manuscript, as they are currently not backed up by sufficient data. As we did not anticipate the early onset and severity of irAEs in this study, we did not have ethical permission to obtain tissue biopsies and had thus no way to investigate this hypothesis in greater detail.

30) Fundamentally, they do not find any significant effects on intra-tumoral Treg numbers in PI3Kdelta inhibitor treated patients vs control. While there is a small decrease in Foxp3 mRNA on bulk TCRseq, there is NO statistical difference by Foxp3 IHC, which is a much better indicator of Treg because there are many post-transcriptional mechanisms that regulate Foxp3 protein; thus, protein, not RNA is critical. I was surprised they did not do Foxp3 by FACS, given that one gets a lot of T cells out of a HNSCC resection and even out of pre-treatment biopsies if one can get a few passes with an 18 guage biopsy needle. They do find an increase in Treg in the peripheral blood but unclear what this means given the absence of significant Treg in the tumor. They conclude that the drug pushes Treg out of the tissue and into the peripheral blood – the dynamics of leukocytes between tissues and blood is complicated and one cannot draw conclusions from simply finding changes in number within a given compartment without employing some sort of tracking approach, which is often done in mouse models but is very difficult in humans.

We thank the reviewer for this comment. While we agree that FACS would have been a better approach to verify a decrease in intratumoral T_{REG} cells, the low TIL infiltration in all tested samples precluded a reliable assessment of this. We assume that a likely reason for this is that HPV-negative HNSCC exhibits an overall lower lower tumor infiltrating lymphocyte (TIL) infiltration, as mentioned in the manuscript. While the described effects on tissue retention of T_{REG} cells are in accordance with previously published studies, we concur that we can't make reliable statements on this without utilizing different approaches like the proposed fate tracking. We have therefore removed these data and pertaining sections from the manuscript and instead focused more heavily on the emergence of irAEs.

31) The analysis of transcriptional programs of Treg, in particular genes known to regulate Treg function, and genes that the authors indicate are regulated by PI3K, are not evaluated. The Treg cells are not identified in the sc analysis in Fig 6a and the violin plots in Fig 6b (they refer to Fig 5b in the text but I think this is a mistake) only look at genes associated with CTL function. Visually, there does not appear to be much of a difference with inhibitor vs control and I did not see any statistics – they also refer to supplementary table 5 but that table is TCRseq. Figure 6c is incomprehensible and 6d doesn't tell us much. Without knowing the specificity of the TCRs, hard to know what the modest clonal expansions ultimately mean.

We thank the reviewer for raising this point. With regard to T_{REG} cells in the scRNA-seq analysis, from the initially sequenced 2,342 CD3⁺ T cells, only 119 FOXP3⁺ cells remained post QC (n=0-27 cells per patient), precluding a more detailed analysis of this cell type. We have to note that the data in question are indeed shown in Supplementary Table 5 in the original version of the manuscript, which has 3 tabs, describing identified differentially expressed genes in CD4⁺ T cells (first tab), CD8⁺ T cells (second tab) and the pertaining TCR data (third tab). Given that the depicted genes were already identified as DEGs, no further statistical validation on this

is needed. We apologize if that was not already stated accordingly. We have moreover de-emphasized these results, as well as the modest increase in clonal expansion and moved these data to the supplement (**Extended Data Fig. 3 a-c**).

In summary, we have thoroughly addressed all of the concerns and comments of the reviewers, as outlined in detail.

Reviewer Reports on the First Revision:

Referees' comments:

Referee #1 (Remarks to the Author):

The authors have overall responded well to the reviewers and markedly improved the manuscript. With respect to the discussion about solid tumor patients being at higher risk of irAEs, it is worth also mentioning the possible effect of age. What was the median age of the patients and were the patients with severe toxicity or who had to stop therapy younger? In B cell malignancies, it's clear that the irAEs are much more likely in younger patients as well as in those without prior therapy so age should also be mentioned as a possible component of the higher risk of irAEs seen here.

In Fig 1, grade 3-4 tox should be the same color (instead of having skin the same color regardless of grade).

There doesn't really seem to be any trend in the IHC cell counts or FOXP3 in Ext data 2. This comment should be modified in the text.

Bottom of p. 7, they should state how they verified that the observed anti-tumor effects were due to the infiltrating immune cells, in the text, as they do in the response to reviewers

Cluster 8 in the mouse colonic Treg analysis – the new text says they show substantial clonal expansion but it is also depleted in PI3Kd treated mice. This needs to be clarified or better explained

The new data on p. 11, I think a little more explanation of which cluster is which would make it easier for the reader.

Referee #2 (Remarks to the Author):

The authors have addressed my comments. I really appreciate that authors not only report an observation but tried to find solution by modifying the schedule.

I still have some minor points, mostly about formatting

the three patients who presented a PR or CR also had a grade 3/4 AE. it would be worth mentioning it somewhere.

the patient who presented a PR in the placebo group is not highlighted in the figure

page 6; it's unclear why the experiments were done with 2 pairs for the sorted CD8 and 6 with the CD3. no need to add experiment, but some clarification would be welcome

in the figure 1; the colors are countintuitive because blue and red do not mean the same for skin AE versus GI AE. authors should homogenize the meaning of the colors.

in the discussion, as the authors state ("presumably cause..."), there is no evidence that it's the Treg depletion that cause pathogenic T cell reaction. This would deserve a formal sentence in the discussion (no need for new experiments). The paper shows that targeting PI3Kd decrease Treg and that this is associated with toxicity and expansion of TH17 response.

The only issue that is remaining from the perspective of the reviewer is the lack of evidence that Tregs decrease in the tumor following PI3Kd inh based on IHC since this is the typical data that is easy to visualize and interpret. This could be explained by the long interval between end of

treatment and surgical procedure. this data should be explained somewhere in the discussion as a limitation. maybe authors should have a specific look at samples obtained at surgery only in patients for which the PI3Kd inh was not stopped prematurely (to avoid long interval between EOT and surgery). In general, the ED Fig 2 is very important to support the whole story and I would recommend moving the panels in the Figure 1.

Referee #2's comments on Referee #3's concerns:

I have read the revised paper and the authors addressed the comments of the reviewer 3, with the exception of the comments 30 and 31 that all relate to the evidence that the PI3Kd inhibitor has an impact of intratumoral RegT cell. Showing that the therapy increases quantity and/or activation of reg T cells in the tumor is an important data for this paper. If the authors can provide such convincing data, the paper could be published.

Author Rebuttals to First Revision:

We are grateful to the referees for their further review of our manuscript and for the very positive assessment. In particular we would like to extend our gratitude to reviewer #2, for also evaluating our manuscript against the points raised by reviewer #3.

We have addressed all comments and provide a point-by-point reply (in blue coloured text) to the reviewer's critiques is provide below. The manuscript and figures have been formatted to the journal requirements.

Point-by-point plan for addressing reviewer's comments (authors responses and changes to manuscript in blue font).

Referee #1 (Remarks to the Author):

1. The authors have overall responded well to the reviewers and markedly improved the manuscript.

We thank the reviewer for this very positive assessment of our work.

2. With respect to the discussion about solid tumor patients being at higher risk of irAEs, it is worth also mentioning the possible effect of age. What was the median age of the patients and were the patients with severe toxicity or who had to stop therapy younger? In B cell malignancies, it's clear that the irAEs are much more likely in younger patients as well as in those without prior therapy so age should also be mentioned as a possible component of the higher risk of irAEs seen here.

We thank the reviewer for raising this question. The median age of patients in our study was 65.5 years. In the 21 patients on drug, the median age was 65.0 years. In the group of patients with grade 3/4 irAE the median age was 59 years, compared to the group of patients who did not stop experience grade 3/4 irAEs with a median age of 66 years. The difference was not significant (Fig. R1).

Fig.R1: Age distribution in PI3Kδi-treated patients that did or did not experience grade 3/4 irAEs

3. In Fig 1, grade 3-4 tox should be the same color (instead of having skin the same color regardless of grade).

This colour scheme has now been adjusted. All grade 3/4 irAEs are now highlighted in red.

4. There doesn't really seem to be any trend in the IHC cell counts or FOXP3 in Ext data 2. This comment should be modified in the text.

As suggested by Reviewer 2, we have now considered the interval between stopping of treatment and IHC assessment of intratumoral T_{REG} cell numbers. Please also see our response to #14.

“As PI3Kδ-inhibition led to a significant reduction in FOXP3 transcript levels in the tumor samples, we assessed T_{REG} cell levels in tumor tissue via immunohistochemistry, hypothesizing that the duration between ceasing of treatment and tumor resection might be critical factor influencing T_{REG} cell abundance due to the relatively short half-life of the compound. Indeed, we found significantly reduced intratumoral T_{REG} cells only in patients in which their abundance could be assessed directly after treatment (PI3Kδi short interval) (Fig. 1c), implying that T_{REG} cell levels normalize quickly once treatment has been stopped.”

5. Bottom of p. 7, they should state how they verified that the observed anti-tumor effects were due to the infiltrating immune cells, in the text, as they do in the response to reviewers

This information has now been added.

“Given that PI3K inhibitors were initially considered to mainly target cancer cell-intrinsic PI3K activity, we utilized RAG1^{-/-} and CD8^{-/-} mice to verify that the observed anti-tumor effects were dependent on immune cells, and more specifically on CD8⁺ T cells (Extended Data Fig. 4h).”

6. Cluster 8 in the mouse colonic Treg analysis – the new text says they show substantial clonal expansion but it is also depleted in PI3Kd treated mice. This needs to be clarified or better explained.

We have modified that statement to explain the results better.

“Cluster 8 colonic T_{REG} cells, which showed substantial clonal expansion in control-treated mice but were depleted in PI3Kδi-treated mice, resembled the recently described tissue-resident ST2 T_{REG} cells¹⁸⁻²⁰, which are critical for the protection against chronic inflammation and facilitation of tissue repair (Fig. 2a,b and Extended Data Table 2).”

7. The new data on p. 11, I think a little more explanation of which cluster is which would make it easier for the reader.

We thank the reviewer for this suggestion and have now added some further details and color labels for the different clusters to the text.

Referee #2 (Remarks to the Author):

8. The authors have addressed my comments. I really appreciate that authors not only report an observation but tried to find solution by modifying the schedule.

We are delighted that the majority of this reviewer's concerns have been addressed and that they are satisfied with the additional data we provided.

9. I still have some minor points, mostly about formatting the three patients who presented a PR or CR also had a grade 3/4 AE. it would be worth mentioning it somewhere.

This information has now been added.

"Two patients with partial responses (PR) and one with complete pathological response occurred in AMG319-treated patients (Extended Data Fig. 1c), all of whom exhibited grade 3/4 irAEs."

10. the patient who presented a PR in the placebo group is not highlighted in the figure

The details in the main figure are true but the statement in the result section was added incorrectly. We apologize for this inadvertent error, and have now amended that statement in the results section.

11. page 6; it's unclear why the experiments were done with 2 pairs for the sorted CD8 and 6 with the CD3. no need to add experiment, but some clarification would be welcome

Due to the relatively small size of the tumors and the different types of analyses that were performed (immunohistochemistry, whole tumor RNA-seq, bulk RNA-seq and single-cell RNA-seq of purified T cells), we had good quality RNA from paired samples of sorted CD8⁺ T cells only for 2 patients (CD8 bulk RNA-seq), and good quality single-cell transcriptomes of T cells from dispersed biopsies only from 6 patients.

12. in the figure 1; the colors are counterintuitive because blue and red do not mean the same for skin AE versus GI AE. authors should homogenize the meaning of the colors.

We have now changed the colors so that all grade 3/4 irAEs are highlighted in red.

13. in the discussion, as the authors state ("presumably cause..."), there is no evidence that it's the Treg depletion that cause pathogenic T cell reaction. This would deserve a formal sentence in the discussion (no need for new experiments). The paper shows that targeting PI3Kd decrease Treg and that this is associated with toxicity and expansion of TH17 response.

We thank the reviewer for this careful comment and have now added the statement as suggested by the reviewer.

"These treatment-mediated changes, specifically depletion of Il10-expressing ST2 T_{REG} cells is associated with colitis and the expansion of pathogenic T_{C17} and T_{H17} T cell subsets in colonic tissue."

14. The only issue that is remaining from the perspective of the reviewer is the lack of evidence that Tregs decrease in the tumor following PI3Kd inh based on IHC since this is the typical data that is easy to visualize and interpret. This could be explained by the long interval between end of treatment and surgical procedure. this data should be explained somewhere in the discussion as a limitation. maybe authors should have a specific look at samples obtained at surgery only in patients for which the PI3Kd inh was not stopped prematurely (to avoid long interval between EOT and surgery).

We thank the reviewer for the excellent suggestion to analyse T_{REG} depletion in tumor tissue specifically in the patients who did not prematurely stop treatment with PI3Kδ inhibitors. We stratified AMG-319-treated patients based on the interval between prematurely stopping AMG-319-treatment and IHC assessment: >4 days (Long interval, LI) or <1 day (short interval, SI). These new analyses, as the reviewer suspected, reveal a clear link between the treatment interval and reduction in T_{REG} in the tumor tissue. These important data have now been added to the manuscript (Fig. 1c).

“As PI3K δ -inhibition led to a significant reduction in FOXP3 transcript levels in the tumor samples, we assessed T_{REG} cell levels in tumor tissue via immunohistochemistry, hypothesizing that the duration between ceasing of treatment and tumor resection might be critical factor influencing T_{REG} cell abundance due to the relatively short half-life of the compound. Indeed, we found significantly reduced intratumoral T_{REG} cells only in patients in which their abundance could be assessed directly after treatment (PI3K δ short interval) (Fig. 1c), implying that T_{REG} cell levels normalize quickly once treatment has been stopped.”

15. In general, the ED Fig 2 is very important to support the whole story and I would recommend moving the panels in the Figure 1.

As requested, these data have now been moved to Fig.1

16. I have read the revised paper and the authors addressed the comments of the reviewer 3, with the exception of the comments 30 and 31 that all relate to the evidence that the PI3K δ inhibitor has an impact of intratumoral RegT cell. Showing that the therapy increases quantity and/or activation of reg T cells in the tumor is an important data for this paper. If the authors can provide such convincing data, the paper could be published.

We thank the reviewer for this comment and assume they mean “decreases quantity and/or activation”. Thanks to their excellent suggestion to analyse T_{REG} depletion in tumor tissue specifically in the patients who did not prematurely stop PI3K δ inhibitors, we now show a significant reduction in intratumoral T_{REG} following PI3K δ inhibitor treatment (Fig. 1c).

Please also see response to #14.

We hope that we have satisfactorily addressed comments of the reviewers and we have modified the manuscript and figures to conform to the journal guidelines. We hope the manuscript is now acceptable for publication in Nature.

Reviewer Reports on the Second Revision:

Referees' comments:

Referee #1 (Remarks to the Author):

The authors have responded well to all comments

Referee #2 (Remarks to the Author):

I don't have any comment, except that the title of the figure 1 is not appropriate given that authors assess mechanisms of anti tumor activity in several panels.